# Trends in childhood cancer: Incidence and survival analysis over 45 years of SEER data

**Iyad Sultan**[1]*, **Ahmad S. Alfaar**[2], **Yaseen Sultan**[3], **Zeena Salman**[4], **Ibrahim Qaddoumi**[5]

**1** Department of Pediatrics and Artificial Intelligence and Data Innovation Office (AIDI), King Hussein Cancer Center, Amman, Jordan, **2** Department of Ophthalmology, Charité Universitätsmedizin Berlin, Berlin, Germany, **3** Department of Biostatistics, University of Iowa, Iowa City, IA, United States of America, **4** Department of Global Pediatric Medicine, St. Jude Children's Research Hospital, Memphis, TN, United States of America, **5** Departments of Global Pediatric Medicine and Oncology, St. Jude Children's Research Hospital, Memphis, TN, United States of America

* isultan@khcc.jo

**Data Availability Statement:** The data utilized in this study are available from the SEER (Surveillance, Epidemiology, and End Results) registry. Researchers can request access to the

## Abstract

### Background

The SEER Registry contains U.S. cancer statistics. To assess trends in incidence and survival and the impact of demographic factors among pediatric patients with cancer, we assessed nearly 5 decades (1975–2019) of data.

### Methods

All patients below the age of 20 with histology-confirmed malignancy were studied. Kaplan-Meier survival curves were generated to evaluate survival trends across treatment periods and ICCC classes. JoinPoint analysis was conducted to identify changes in incidence and survival.

### Results

The incidence of childhood cancer increased from 14.23 cases per 100,000 children in 1975–1979 to 18.89 in 2010–2019, with an average annual percent change of 0.73. This rise was more pronounced in several cancers, including leukemias, lymphomas, brain tumors, hepatic tumors, and gonadal germ cell tumors. Age-adjusted cancer mortality decreased from 4.9 to 2.3 per 100,000. Cancer-related mortality was consistently higher in boys than in girls, and in Black children than in White children. Survival significantly improved, with 5- and 10-year survival rates rising from 63.1% to 85.2% and from 58.8% to 82.7%, respectively. Leukemias showed a substantial increase in 5-year survival from 48.2% ± 1.7% to 85.1% ± 0.4% in 2010–2019. Lymphomas also showed significant improvement, with survival increasing from 72.9% ± 1.7% to 94.2% ± 0.3%. Despite these improvements, the survival of CNS tumors, bone tumors, and sarcomas remained suboptimal, with 5-year survival estimates of approximately 60%. Our joinpoint analysis confirmed our findings but revealed an interesting increase in the incidence of lymphomas limited to the years between 2005 and 2014.

data directly from the SEER registry: https://seer.cancer.gov/.

**Funding:** The author(s) received no specific funding for this work.

**Competing interests:** The authors have declared that no competing interests exist.

## Conclusion

This research elucidates advancements in survival among pediatric patients with cancer. The results offer critical perspectives on pediatric oncology, highlighting the imperative for ongoing innovation in therapeutics. Although the increase in incidence may partially stem from enhanced diagnostic capabilities and more comprehensive registration processes, the underlying causes remain unclear.

## Introduction

Childhood cancer remains a significant public health concern worldwide, despite substantial progress in its management over the last 5 decades. Advances in diagnostic tests, risk stratification, and therapeutic interventions have improved survival of many pediatric malignancies [1]. However, the incidence of specific cancer types and disparities in outcomes among different population groups persist [2]. Additionally, long-term sequelae affect the quality of life of cancer survivors, emphasizing the need for further research to minimize treatment-related toxicities [3].

Assessing trends in childhood cancer incidence, survival, and mortality is crucial to understanding the effectiveness of current interventions and identifying areas where additional efforts are needed. Monitoring these trends can also help to identify potential risk factors, allocate healthcare resources, and guide public health policies. Furthermore, understanding the disparities in outcomes across different populations will enable us to better address the inequity in cancer care and in the implementation of targeted interventions.

Over the last fifty years, the landscape of pediatric oncology has undergone significant evolution, marked by the introduction and refinement of chemotherapy, spearheaded by collaborative groups across North America and Europe. These efforts have led to the development of more effective treatment regimens that optimize the use of established drugs, resulting in markedly improved outcomes for almost all types of pediatric cancer. Enhancements in supportive care have rendered intensive treatments more manageable. Advances in stem cell transplantation techniques have become pivotal in rescuing patients who do not respond to initial treatments. Diagnostic progress, including molecular stratification, detection of minimal residual disease, and sophisticated genetic profiling, has refined therapeutic approaches, allowing for more tailored and effective treatments. Improvements in imaging technologies, such as advanced CT scanners and the advent of nuclear scanning, have significantly improved the detection of metastatic disease. Surgical and radiation oncology techniques have also seen substantial advancements, improving the precision and efficacy of tumor resection and control. The introduction of targeted therapies and immunotherapies has opened new avenues for treating specific patient subsets, including those with acute lymphoblastic leukemia (ALL), high-risk neuroblastoma, relapsed Hodgkin lymphoma, and others, marking a shift towards precision medicine. The integration of multidisciplinary care teams has further optimized treatment outcomes and patient care, emphasizing the importance of a holistic approach in the management of pediatric cancers [4, 5].

The Surveillance, Epidemiology, and End Results (SEER) registry, a comprehensive source of cancer statistics in the United States, provides a unique opportunity to examine the changing trends and mortality rates in childhood cancer over an extended period [6]. By analyzing data from the SEER registry, we can assess the impact of advancements in diagnostics and therapeutics on the epidemiology and outcomes of pediatric malignancies.

This study aimed to provide a comprehensive overview of nearly 5 decades of childhood cancer statistics by examining trends and mortality rates in the SEER registry and providing insight into the progress and challenges ahead. We examined major pediatric cancer types, survival, and mortality rates, stratified by sex, age, and race/ethnicity. Our findings provide valuable insights into the progress made diagnostics, therapeutics, and clinical management. They also highlight areas in which further research and development are needed to improve outcomes, reduce treatment-related toxicities, and ensure equitable cancer care for all children and adolescents.

## Methods

### Study population and definitions

A retrospective analysis was conducted using data from the SEER database, encompassing the period 1975–2019. All children and adolescents (aged 0–19 years; i.e. below the age of 20) with cancer during this time frame were included. We have conducted two main analyses; population based for incidence rates, relative survival and mortality, and we generated a cohort-analysis for general characteristics and overall survival of this cohort. For the population cohort, we depended on the SEER 8 database. Data were extracted using SEER*stat [7] by using the rates, case-listing and survival sessions. For the rates and relative survival sessions, SEER 8 (Nov 2021 submission, 1975–2019) was employed, and for the case-listing session, a data set of SEER 8 (1975–2019), SEER 12 (1992–2019), and SEER 17 (2000–2019) was constructed. The datasets were merged and duplicates were removed based on the first primary label and patient identifier. The International Classification of Childhood Cancer (ICCC) site/histology recode (ICCC Site Recode Extended 3rd Edition/IARC 2017) [8] was utilized to label all cancers. Race, sex, and survival data were also extracted. For all sessions, the study durations were stratified as follows: 1975–1979, 1980–1989, 1990–1999, 2000–2009, and 2010–2019. These periods were employed as user-defined variables. Age-standardized incidence rates (ASIRs) for each cancer type were obtained to assess incidence trends using the direct method and the 2000 U.S. standard population as a reference. Incidence rates were stratified by sex, age group (<1, 1–4, 5–9, 10–14, and 15–19 years), and race/ethnicity (White, Black, other, and unknown). Average percentage change (APC) and $p$-values for change over time (1975–2019) were calculated and provided by SEER*stat.

### Ethical approval

Ethical approval was not required for this study. The Institutional Review Board (IRB) of King Hussein Cancer Center (KHCC) waived the need for ethical approval as the study posed minimal risk and utilized securely de-identified data.

### Statistical methods

Survival was determined using the Kaplan-Meier method. We calculated the probability of 5-year overall survival (OS) for each cancer type and stratified the results by sex, age group, and race/ethnicity. Trends in survival over the study period were also assessed. To compare the survival of different subgroups across decades, Cox regression was applied, and the $p$-values were adjusted using the Holm method (also known as the Holm-Bonferroni method), a stepwise multiple-testing correction used to control the family-wise error rate when performing multiple hypothesis tests. The family-wise error rate is the probability of making at least one false-positive (Type I) error among all the hypothesis tests performed. This adjustment was necessary due to the large sample size and the fact that multiple testing can yield

significant p-values by chance. Variables included in our multivariable model were: age, recoded in five-year increments; race; sex; SEER stage; decade, representing the time period or year of diagnosis in ten-year increments; and the ICCC.

This study rigorously assessed the proportional hazards assumption. We evaluated each covariate within the model for proportional hazards over time, using pairs of decades as categorical predictors alongside clinical variables such as diagnosis type. To assess multicollinearity among the predictors, we computed the Variance Inflation Factors (VIF) for each variable using [insert software/tools]. VIF values below the threshold of 5 were considered acceptable, indicating low multicollinearity. Additionally, we computed Cramér's V statistics for categorical variables to evaluate associations between the variables. The VIF values for all variables were below 5, with race and SEER stage exhibiting particularly low multicollinearity (VIF < 1.5), indicating that multicollinearity was not a concern in our model. Cramér's V statistics showed weak associations among the categorical variables, further supporting the absence of multicollinearity concerns.

Descriptive statistics on extracted individual patients were used to summarize the demographic and clinical characteristics of the study population. Continuous variables were reported as means and standard deviations (SD), and categorical variables were presented as frequencies and percentages. Statistical significance was set at $p <0.05$, and all tests were two-sided.

Age-standardized incidence rates (ASIRs) for each cancer type were obtained to assess incidence trends using the direct method and the 2000 U.S. standard population as a reference. Incidence rates were stratified by sex, age group (<1, 1–4, 5–9, 10–14, and 15–19 years), and race/ethnicity (White, Black, other, and unknown). Average percentage change (APC), standard errors and $p$-values for change over time (1975–2019) were calculated All analyses were conducted using SEER*Stat and R software (version 4.2.0). The study was deemed exempt from institutional review board approval, as the SEER database contains de-identified data and posed minimal risk to individual privacy.

## JoinPoint regression

The JoinPoint Regression program (version 4.9.1.0) was employed to assess age-standardized incidence trends by fitting the most straightforward joinpoint model to cancer annual rate data. This analysis aimed to detect significant alterations in trends and determine the significance of apparent changes using the Monte Carlo Permutation method, assuming constant variance and uncorrelated errors [9]. JoinPoint Regression for incidence was conducted on annual age-adjusted rates from SEER 8 registries. We calculated the APC for the whole group and the slope values for the smaller groups (cancer groups according to the ICCC) to provide a more sensitive method to assess changes and fluctuations. Further explanation of the methods can be found elsewhere [10]. Relative survival was used to calculate the net survival in the absence of other causes of death, and the Ederer II method was used to calculate the cumulative expected survival. The U.S. 1970–2018 expected survival table, by individual year and race (White/Black/other), was used to calculate relative survival. To analyze relative survival trends based on the year of diagnosis, JPSurv online software (accessed on April 30, 2023) was utilized, which applies JoinPoint survival models to identify shifts in linear trends in cancer death hazards over time [11–13]. The joinpoint survival model is a type of proportional hazard model for survival that extends its functionality to include the effect of the calendar year at diagnosis on the log hazard scale of cancer death. In this model, the effect of the calendar year is assumed to be linear. A maximum of three joinpoints were allowed for the analysis. Average absolute change in survival (AACS) was calculated between the survival joinpoints.

## Results

### Changes in rates

Incidence rates steadily increased over time, starting with 14.23 cases per 100,000 children in 1975–1979 and rising to 15.24 in 1980–1989, 15.98 in 1990–1999, 17.25 in 2000–2009, and 18.89 in 2010–2019. The average annual percentage change (APC) in the incidence of all childhood cancers over the study period was 0.73 for all patients included. The reported changes for different racial groups revealed that Black children had the lowest rate of change (APC 0.48), while White (APC 0.78) and children of other races (APC 0.71) exhibited higher rates of change.

We also found significant increases in the incidence rates of leukemias, myeloproliferative, and myelodysplastic diseases (APC 0.84); lymphomas and reticuloendothelial neoplasms (APC 0.72); CNS and miscellaneous intracranial and intraspinal neoplasms (APC 0.71); soft-tissue and other extraosseous sarcomas (APC 0.43); hepatic tumors (APC 2.17); and germ cell tumors, trophoblastic tumors, and neoplasms of gonads (APC 0.50) (Table 1 and Fig 1). In contrast, the incidence rates of other cancer types, such as neuroblastoma and other peripheral nervous cell tumors (APC 0.14), retinoblastoma (APC 0.21), renal tumors (APC -0.18), and malignant bone tumors (APC 0.23) were relatively stable.

The increase in leukemias was driven mainly by an increase in precursor cell leukemias (APC, 0.64), which was highest for White females (APC, 0.83), acute myeloid leukemias (AML; APC, 0.88), and chronic myeloproliferative diseases (APC, 2.42). The increase in lymphomas was mainly due to mature non-Burkitt B-cell lymphomas (APC, 0.93). The increase in CNS tumors was mainly due to the incidence of ependymomas (APC, 0.76), which was highest for White females (APC, 1.28); astrocytomas (APC, 0.66), highest for White females (APC, 0.77); and mixed and unspecified gliomas (APC, 1.44). Hepatoblastoma in both sexes and all races had an APC of 2.31. The increased APC of germ cell tumors was led by an increase in malignant gonadal germinomas in the White race (APC, 1.73). Melanoma incidence increased significantly in the White race (APC, 0.73, p = 0.01). A significant increase in the incidence of osteosarcoma in the White race was seen (APC, 0.86), compared to a nonsignificant decrease in the Black race (APC, -0.63; p = 0.23). The incidence of skeletal Ewing sarcoma was steady (APC, -0.22).

**Table 1.  Decadal Incidence rates of childhood cancer across all ICCC categories in the SEER 8 data set, expressed per 100,000.**

| ICCC Categories | Study Intervals | | | | | APC [CI] | p-value |
|---|---|---|---|---|---|---|---|
| | 1975–1978 | 1980–1989 | 1990–1999 | 2000–2009 | 2010–2019 | | |
| I Leukemias, myeloproliferative and myelodysplastic diseases | 3.38 | 3.72 | 3.9 | 4.43 | 4.66 | 0.84 [0.68, 1.00] | <0.01 |
| II Lymphomas and reticuloendothelial neoplasms | 2.41 | 2.48 | 2.35 | 2.43 | 3.2 | 0.72 [0.45, 0.98] | <0.01 |
| III CNS and miscellaneous intracranial and intraspinal neoplasms | 2.36 | 2.59 | 2.86 | 3.13 | 3.15 | 0.71 [0.45, 0.97] | <0.01 |
| IV Neuroblastoma and other peripheral nervous cell tumors | 0.82 | 0.82 | 0.82 | 0.87 | 0.85 | 0.14 [-0.18, 0.47] | 0.38 |
| IX Soft-tissue and other extraosseous sarcomas | 1.04 | 1.08 | 1.14 | 1.23 | 1.22 | 0.43 [0.08, 0.78] | 0.02 |
| V Retinoblastoma | 0.28 | 0.28 | 0.34 | 0.3 | 0.32 | 0.21 [-0.30, 0.72] | 0.41 |
| VI Renal tumors | 0.64 | 0.69 | 0.68 | 0.64 | 0.62 | -0.18 [-0.60, 0.24] | 0.39 |
| VII Hepatic tumors | 0.13 | 0.15 | 0.21 | 0.26 | 0.31 | 2.17 [1.45, 2.89] | <0.01 |
| VIII Malignant bone tumors | 0.77 | 0.91 | 0.91 | 0.87 | 0.92 | 0.23 [-0.10, 0.57] | 0.16 |
| X Germ cell tumors, trophoblastic tumors, and neoplasms of the gonads | 0.86 | 1.04 | 1.1 | 1.16 | 1.14 | 0.50 [0.19, 0.81] | <0.01 |

Abbreviations: APC, annual percent change; CI, confidence interval; CNS, central nervous system; ICCC, International Childhood Cancer Classification

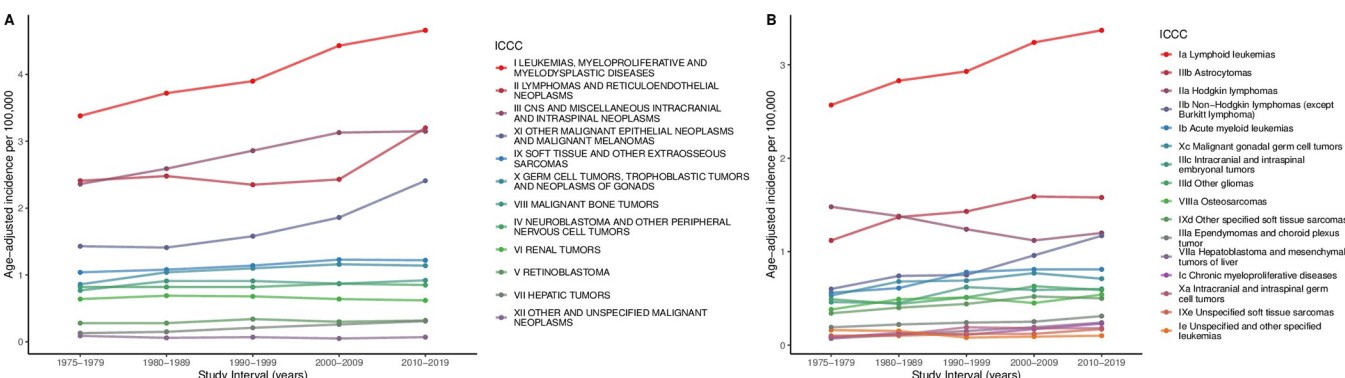

**Fig 1.** Trends in age-standardized incidence rates showing the trends of (**A**) ICCC classes and (**B**) the 20 most common cancers in pediatric patients.

Male and female patients across all racial categories had an increased cancer rate over the analyzed periods. For female patients of all races, the incidence increased from 13.82 in 1975–1979 to 18.15 in 2010–2019, with an APC of 0.73. Higher trends were observed for females of White race (APC 0.80) and females of other races (APC 0.75), in comparison to those of Black race (APC 0.22). Likewise, for male patients of all races, the incidence increased from 14.63 in the 1975–1979 to 19.61 in 2010–2019, with an APC of 0.72. The trend was consistent across racial groups, with Black males (APC 0.68), other males (APC 0.67), and White males (APC 0.75) also experiencing increased incidence rates. Full details of rate trends are provided in S1 Table.

**Age-adjusted all-cause mortality rates.** Mortality records showed significant reductions in mortality rates for all children across various demographics over the analyzed period (Fig 2). The age-adjusted all-cause mortality rates for all children declined from 125.0 per 100,000 in 1975–1979 to 52.0 per 100,000 in 2010–2019, indicating a substantial improvement in child health outcomes. A similar trend was observed for cancer as a cause of death among children, with the age-adjusted death rate dropping from 4.9 per 100,000 in 1975–1979 to 2.3 per 100,000 in 2010–2019. When stratified by sex, male patients consistently exhibited higher age-adjusted cancer mortality rates than did female patients. The highest age-adjusted cancer mortality rates were consistently observed in the 15–19 years age group, followed by the 5–9 years and 10–14 years age groups.

## Cancer survival trends across study periods

Kaplan-Meier survival estimates revealed significant improvements in pediatric 5- and 10-year survival. Rates rose from 63.1% ± 0.8% and 58.8% ± 0.8% in 1975–1979 to 85.2% ± 0.2% and 82.7% ± 0.3% in 2010–2019 (Fig 3).

The survival of each disease generally improved, with significant improvements noted between consecutive decades for most diseases (Table 2 and Fig 4). Leukemias showed a substantial increase in survival, from 48.2% ± 1.7% in 1975–1979 to 85.1% ± 0.4% in 2010–2019. Lymphoma improved from 72.9% ± 1.7% to 94.2% ± 0.3%. CNS and miscellaneous intracranial and intraspinal neoplasms showed improved survival from 58.6% ± 2% to 74.6% ± 0.6%. These improvements were observed consistently across sex, race, and age groups. The data also showed that some diseases (e.g., retinoblastoma) already had high survival (95.8% ± 2.4%) during the earliest study period, and insignificant changes (95.6% ± 0.9%) by the latest decade. Throughout the study period, black children did worse than white children.

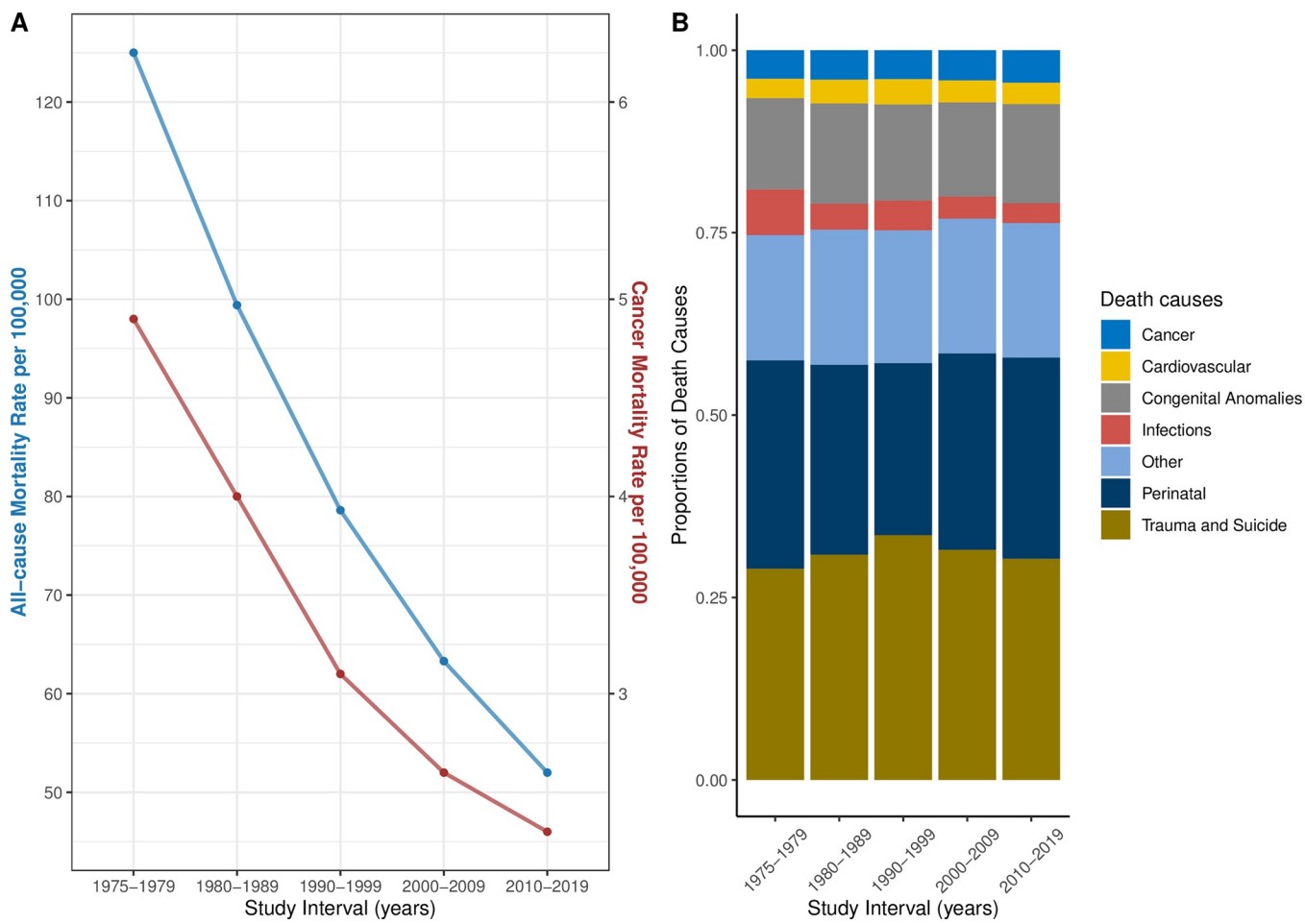

**Fig 2.** (**A**) Trends in age-adjusted death rates are shown for all-cause mortality (blue) and cancer-related mortality (yellow). (**B**) The distribution of causes of death in children is stratified by study interval.

## Survival of the most common types of cancer

Survival of various cancers has generally improved over time, with significant improvements noted between consecutive study periods for several diseases (Table 3 and Fig 5). Precursor cell leukemias (Ia1) demonstrated a significant increase in survival from 55.8% ± 1.9% in 1975–1979 to 89.1% ± 0.4% in 2010–2019, with *p*-values <0.001 across all periods. Similarly, acute myeloid leukemias (Ib) significantly improved from 22.5% ± 3.3% to 67.5% ± 1.2%, with most decade-wise comparisons yielding p-values <0.001. High survival was noted for Hodgkin lymphomas (IIa) at 86.6% ± 1.6% in 1975–1979, which increased to 97.6% ± 0.3% in 2010–2019.

Astrocytomas (IIIb) exhibited a significant increase from 68.5% ± 2.7% to 81.5% ± 1.1% (p <0.001) between 1980–1989 and 1990–1999. Subsequently, the rates remained relatively stable, with no significant differences between consecutive intervals. Medulloblastomas (IIIc1) demonstrated improved survival from 49.2% ± 4.6% in 1975–1979 to 76.3% ± 1.6% in 2010–2019, with remarkable improvements after 1980 (p = 0.032) and 2010 (p = 0.029). In contrast, mixed and unspecified gliomas (IIId2) showed no significant changes in survival between subsequent periods, with rates of 41.2% ± 5.3% in 1975–1979 and 57.6% ± 1.7% in 2010–2019.

We found notable improvements in the survival of patients with neuroblastoma or ganglioneuroblastoma (IVa) over the past four decades. The probability increased from 51.8% ± 3.5%

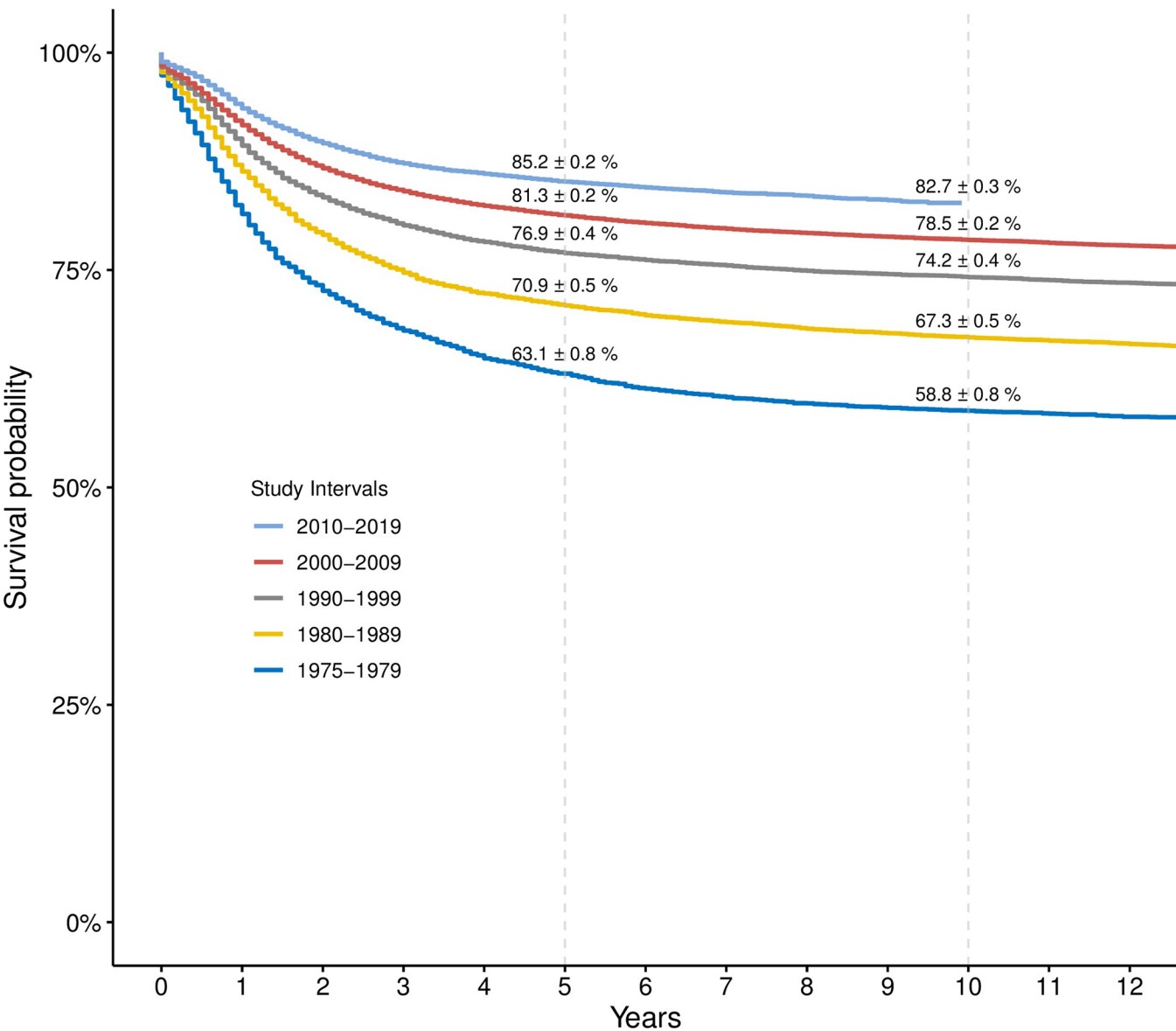

**Fig 3. Kaplan-Meier survival curves for all pediatric patients with cancer, stratified by the study interval during which the diagnosis was made.**

in 1975–1979 to 54.7% ± 2.3% in 1980–1989, with no significant change between these periods (p >0.05). Subsequently, significant increases were recorded in 1990–1999 (68.1% ± 1.8%, p <0.001), 2000–2009 (75.8% ± 1%, p <0.001), and 2010–2019 (80.8% ± 1.1%, p <0.001), highlighting the considerable progress in patient outcomes over time.

Rhabdomyosarcomas (IXa) displayed significant improvement only from 1975–1979 (47.9% ± 4.5%) to 1980–1989 (63.3% ± 3.2%, p = 0.007), but no significant change occurred afterwards. Nephroblastoma (VIa1) revealed a significant increase in survival from 76% ± 3.5% in 1975–1979 to 94.4% ± 0.8% in 2010–2019 (p = 0.024). Osteosarcomas (VIIIa) showed a slow, nonsignificant improvement in survival over the past four decades from 58.8% ± 3.1% in 1980–1989 to 68.3% ± 1.6% in 2010–2019. Thyroid carcinomas (XIb) maintained high and stable survival at 99.3% ± 0.7% in 1975–1979 and 99.5% ± 0.2% in 2010–2019.

**Table 2. Kaplan-Meier survival estimates per study interval, with inter-decadal comparisons made using Cox regression analysis.**

| ICCC Categories | 1975–1979[1] | 1980–1989[1] | p-value[2] | 1990–1999[1] | p-value[2] | 2000–2009[1] | p-value[2] | 2010–2019[1] | p-value[2] |
|---|---|---|---|---|---|---|---|---|---|
| I Leukemias, myeloproliferative and myelodysplastic diseases | 48.2 ± 1.7 | 62 ± 1.1 | <0.001 | 72.8 ± 0.7 | <0.001 | 80.6 ± 0.4 | <0.001 | 85.1 ± 0.4 | <0.001 |
| II Lymphomas and reticuloendothelial neoplasms | 72.9 ± 1.7 | 78.5 ± 1.1 | <0.001 | 86.2 ± 0.8 | <0.001 | 89.8 ± 0.4 | 0.001 | 94.2 ± 0.3 | <0.001 |
| III CNS and miscellaneous intracranial and intraspinal neoplasms | 58.6 ± 2 | 64.6 ± 1.3 | 0.011 | 69.9 ± 0.9 | <0.001 | 72.7 ± 0.5 | 0.024 | 74.6 ± 0.6 | 0.006 |
| IV Neuroblastoma and other peripheral nervous cell tumors | 52 ± 3.5 | 55.1 ± 2.3 | 0.392 | 68.5 ± 1.7 | <0.001 | 76 ± 1 | <0.001 | 81 ± 1.1 | <0.001 |
| IX Soft-tissue and other extraosseous sarcomas | 64.4±2.8 | 72.3 ± 1.9 | 0.011 | 73 ± 1.4 | >0.9 | 71.8 ± 0.9 | >0.9 | 73.9 ± 1 | 0.347 |
| V Retinoblastoma | 95.8±2.4 | 93.4 ± 1.9 | >0.9 | 96 ± 1.1 | >0.9 | 97.1 ± 0.6 | >0.9 | 95.6 ± 0.9 | >0.9 |
| VI Renal tumors | 72.7±3.5 | 88.1 ± 1.6 | <0.001 | 86.9 ± 1.4 | 0.395 | 89.2 ± 0.8 | 0.33 | 91.7 ± 0.8 | 0.159 |
| VII Hepatic tumors | 23.5±7.3 | 49.5 ± 5.4 | 0.013 | 52.9 ± 3.6 | 0.356 | 65.9 ± 2 | 0.016 | 77.1 ± 1.9 | <0.001 |
| VIII Malignant bone tumors | 49.3 ±3.3 | 56.9 ± 2.3 | 0.198 | 64.2 ± 1.8 | 0.001 | 68.8 ± 1 | 0.198 | 71.5 ± 1.2 | 0.052 |
| X Germ cell tumors, trophoblastic tumors, and neoplasms of gonads | 71.3 ± 2.8 | 84 ± 1.5 | 0.001 | 89 ± 1 | 0.039 | 90.6 ± 0.6 | 0.039 | 92.3 ± 0.6 | 0.013 |
| XI Other malignant epithelial neoplasms and malignant melanomas | 86.9 ± 1.6 | 89 ± 1.1 | 0.089 | 89.3 ± 0.9 | >0.9 | 91.9 ± 0.4 | 0.004 | 94.5 ± 0.4 | <0.001 |
| XII Other and unspecified malignant neoplasms | 40 ± 9.8 | 67.8 ± 8.4 | 0.153 | 71.6 ± 6.4 | >0.9 | 81.1 ± 3.3 | 0.359 | 82.6 ± 3.6 | >0.9 |
| **Survival According to Demographics** | | | | | | | | | |
| Sex | | | | | | | | | |
| *Female* | 69 ± 1.1 | 74.3 ± 0.7 | <0.001 | 78.6 ± 0.5 | <0.001 | 82.7 ± 0.3 | <0.001 | 86.1 ± 0.3 | <0.001 |
| *Male* | 57.6 ± 1.1 | 68 ± 0.7 | <0.001 | 75.6 ± 0.5 | <0.001 | 80.1 ± 0.3 | <0.001 | 84.4 ± 0.3 | <0.001 |
| Race | | | | | | | | | |
| *Black* | 59.1 ± 2.9 | 62.4 ± 2 | 0.429 | 72.5 ± 1.3 | <0.001 | 74.4 ± 0.7 | 0.429 | 79.9 ± 0.7 | <0.001 |
| *Others* | 63.7 ± 3.1 | 65.9 ± 1.8 | 0.069 | 72.9 ± 1.1 | <0.001 | 78.1 ± 0.7 | 0.001 | 83.5 ± 0.7 | <0.001 |
| *Unknown* | 91.3 ± 5.9 | 72.9 ± 7.7 | 0.792 | 95 ± 2 | 0.003 | 91.2 ± 1.4 | 0.672 | 95.6 ± 0.8 | 0.057 |
| *White* | 63.2 ± 0.8 | 72.2 ± 0.5 | <0.001 | 77.8 ± 0.4 | <0.001 | 82.4 ± 0.2 | <0.001 | 85.9 ± 0.2 | <0.001 |
| Age at diagnosis (years) | | | | | | | | | |
| *00– <01* | 60 ± 3.1 | 66.7 ± 1.9 | 0.205 | 74.4 ± 1.4 | 0.001 | 76.4 ± 0.8 | 0.205 | 80 ± 0.9 | 0.017 |
| *01–04* | 60.1 ± 1.7 | 69.3 ± 1 | <0.001 | 77.7 ± 0.7 | <0.001 | 83.5 ± 0.4 | <0.001 | 86.7 ± 0.4 | <0.001 |
| *05–09* | 60.4 ± 1.9 | 70.3 ± 1.2 | <0.001 | 77.1 ± 0.8 | <0.001 | 81.5 ± 0.5 | <0.001 | 85.6 ± 0.5 | <0.001 |
| *10–14* | 59.7 ± 1.8 | 68.1 ± 1.2 | <0.001 | 76.3 ± 0.8 | <0.001 | 80.2 ± 0.5 | <0.001 | 83.8 ± 0.5 | <0.001 |
| *15–19* | 68.5 ± 1.2 | 75 ± 0.8 | <0.001 | 77.2 ± 0.7 | <0.001 | 81.3 ± 0.4 | <0.001 | 85.8 ± 0.3 | <0.001 |

[1]All survival values are presented as mean overall survival (%) ± SD (%)

[2]Holm-Bonferroni adjusted p-values were calculated by comparing the survival during each study interval with that of the previous interval.

**Factors affecting survival comparing study periods.** From 1975–1979 to 2010–2019, distant SEER stage cases increased from 31% to 42%, while unknown/in situ cases substantially decreased from 40% to 5.5%. Localized stages increased from 20% to 36%, and regional cases increased from 9% to 17% (Fig 6A). The distribution of SEER stages among children with cancer evolved considerably, and cancer types that were not staged initially were assigned to SEER stages during the later study periods. Localized, regional, and unknown/in situ SEER stages showed substantially better survival than distant stages (S2 Table).

Multivariable Cox regression analysis showed improved survival for patients who received a cancer diagnosis in subsequent decades. The hazard ratios (HR) for all-cause mortality significantly decreased, with patients diagnosed in 1980–1989 having an HR of 0.73; 1990–1999, 0.53; 2000–2009, 0.44; and 2010–2019, 0.35; all compared to the HR for 1975–1979. Sex, race, age, and SEER stage also played roles in survival, which dramatically improved for almost all

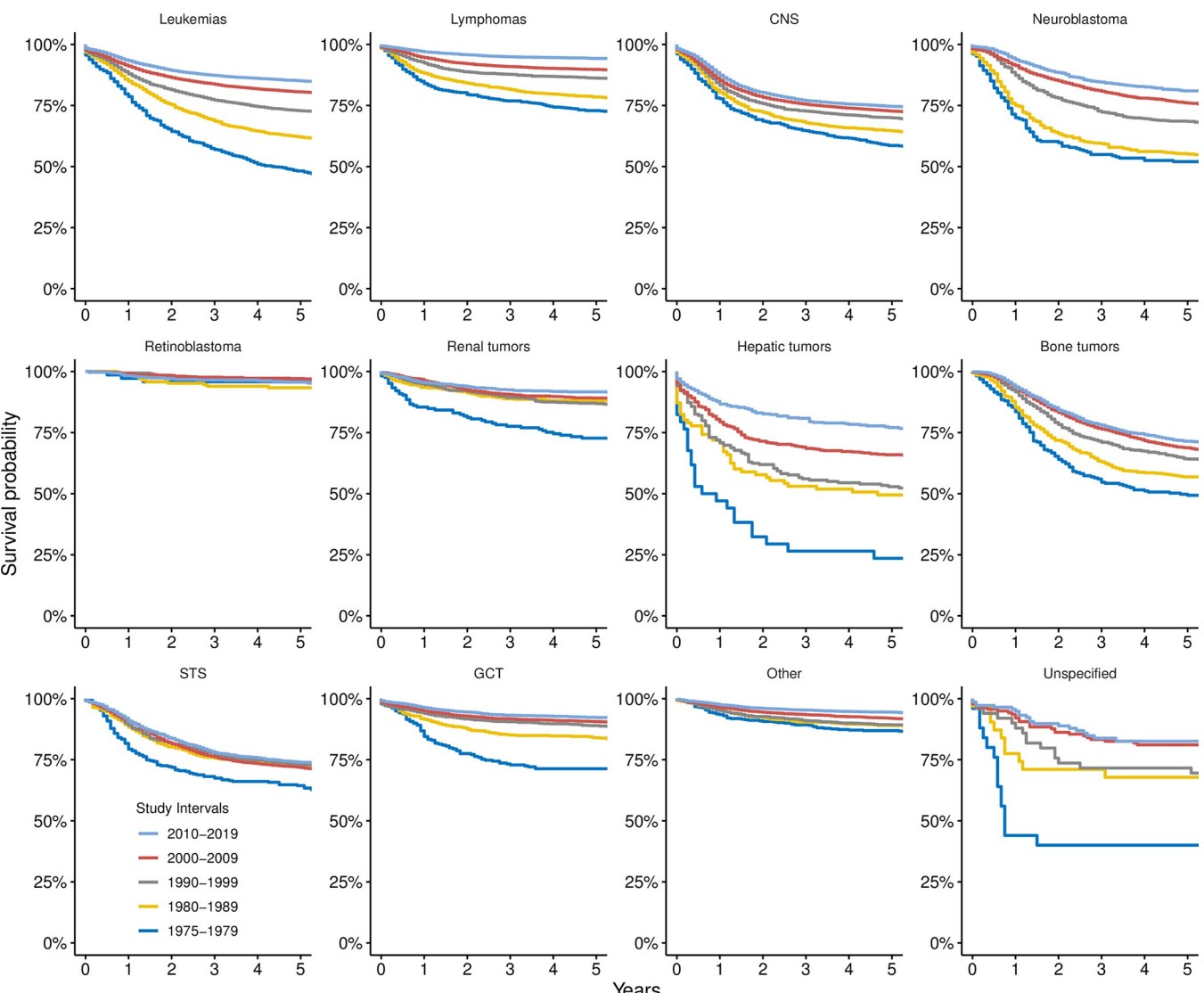

**Fig 4. Kaplan-Meier 5-year survival curves for different ICCC classes, stratified by the study interval during which the diagnosis was made.**
Abbreviations: CNS, central nervous system; GCT, germ cell tumor; STS, soft-tissue sarcoma.

disease categories (Fig 7A). Male patients had a slightly higher HR for death than did female patients (1.14) (Fig 7B). Patients in the Other and White race categories experienced better survival than those in the Black race category (Fig 7C). Patients aged 1 to 14 years at diagnosis (1–4, 5–9, and 10–14 years) showed better survival gains compared to the infant group (0–<1 year) and the oldest (15–19 years) group (Fig 7D).

## JoinPoint analysis

The analysis utilizing JoinPoint trends demonstrated a progressive escalation in the annual incidence of total malignancies, reflected by an APC of 0.73 (p <0.05) (Fig 8). This increase was reflected among all races (Mainly, White: slope = 1.34, Black: slope = 0.63; in both p <0.05). This growth was detected in leukemias (slope = 0.34, p <0.05), CNS malignancies (slope = 0.20, p <0.05), and to a lesser extent in germ cell and hepatic malignancies

**Table 3. Decade-wise 5-year overall survival of the 20 most common childhood cancers and inter-decadal comparisons made using Cox regression analysis.**

| ICCC | 1975–1979 | 1980–1989 | p-value[1] | 1990–1999 | p-value[1] | 2000–2009 | p-value[1] | 2010–2019 | p-value[1] |
|---|---|---|---|---|---|---|---|---|---|
| **Ia1 Precursor cell leukemias** | 55.8%±1.9% | 70.5% ±1.2% | <0.001 | 80.8% ±0.7% | <0.001 | 86.1% ±0.4% | <0.001 | 89.1% ±0.4% | <0.001 |
| Ib AML | 22.5%±3.3% | 32.4% ±2.6% | 0.013 | 44.1% ±1.9% | <0.001 | 59.9% ±1.2% | <0.001 | 67.5% ±1.2% | <0.001 |
| **IIa Hodgkin lymphomas** | 86.6%±1.6% | 89.1% ±1.1% | 0.052 | 94.8% ±0.7% | <0.001 | 95.5% ±0.4% | 0.529 | 97.6% ±0.3% | <0.001 |
| **IIb1 Precursor cell lymphomas** | 50%±15.8% | 65.8%±5% | 0.221 | 74.4% ±3.3% | 0.221 | 81.3% ±1.6% | 0.221 | 85.2% ±1.5% | 0.221 |
| **IIb2 Mature B-cell lymphomas (except Burkitt lymphoma)** | 62.3%±5.8% | 71.2% ±3.4% | 0.28 | 74.2% ±2.6% | 0.306 | 85.1% ±1.3% | 0.005 | 90.9%±1% | <0.001 |
| **IIc Burkitt lymphoma** | 34.1%±7.1% | 55%±4.5% | 0.028 | 80.5% ±2.9% | <0.001 | 87.8% ±1.4% | 0.028 | 93.8% ±1.1% | 0.002 |
| **IId Miscellaneous lymphoreticular neoplasms** | 35.7% ±12.8% | 59.6% ±7.2% | 0.255 | 69%±6.1% | 0.255 | 87.6% ±2.3% | 0.006 | 98.3% ±0.4% | <0.001 |
| **IIIa1 Ependymomas** | 32.7%±6.7% | 53%±4.6% | 0.026 | 67.2% ±3.4% | 0.015 | 73.7% ±1.9% | 0.09 | 84.2% ±1.9% | <0.001 |
| **IIIb Astrocytomas** | 68.5%±2.7% | 71.6% ±1.7% | 0.753 | 81.5% ±1.1% | <0.001 | 83.2% ±0.7% | 0.753 | 81.7% ±0.7% | 0.8 |
| **IIIc1 Medulloblastomas** | 49.2%±4.6% | 62.1% ±3.4% | 0.032 | 67.9% ±2.7% | 0.232 | 70.9% ±1.6% | 0.496 | 76.3% ±1.6% | 0.029 |
| **IIId2 Mixed and unspecified gliomas** | 41.2%±5.3% | 52.5% ±3.6% | 0.214 | 43.6% ±2.9% | 0.523 | 52.2% ±1.6% | 0.09 | 57.6% ±1.7% | 0.065 |
| IVa Neuroblastoma and GNB | 51.8%±3.5% | 54.7% ±2.3% | 0.404 | 68.1% ±1.8% | <0.001 | 75.8%±1% | <0.001 | 80.8% ±1.1% | <0.001 |
| **IXa Rhabdomyosarcomas** | 47.9%±4.5% | 63.3% ±3.2% | 0.007 | 65.7% ±2.4% | >0.9 | 64.1% ±1.5% | >0.9 | 66.1% ±1.8% | 0.802 |
| V Retinoblastoma | 95.8%±2.4% | 93.4% ±1.9% | >0.9 | 96%±1.1% | >0.9 | 97.1% ±0.6% | >0.9 | 95.6% ±0.9% | >0.9 |
| **VIa1 Nephroblastoma** | 76%±3.5% | 89.9% ±1.6% | 0.001 | 89.3% ±1.3% | 0.461 | 91.2% ±0.8% | 0.359 | 94.4% ±0.8% | 0.024 |
| **VIIIa Osteosarcomas** | 42.8%±4.7% | 58.8% ±3.1% | 0.028 | 64.4% ±2.3% | 0.127 | 66.1% ±1.4% | 0.822 | 68.3% ±1.6% | 0.822 |
| **VIIIc1 Ewing tumor and Askin tumor of bone** | 45.9%±5.8% | 49.1% ±3.7% | 0.801 | 58%±3.4% | 0.027 | 68.1% ±1.9% | 0.147 | 71.1% ±2.2% | 0.147 |
| **Xc6 Malignant gonadal tumors of mixed forms** | 100%±0% | 94.1% ±5.7% | >0.9 | 94.5%±2% | >0.9 | 93.4%±1% | >0.9 | 96.1% ±0.8% | 0.248 |
| **XIb Thyroid carcinomas** | 99.3%±0.7% | 99.3% ±0.5% | 0.348 | 98.4% ±0.6% | 0.607 | 98.8% ±0.3% | >0.9 | 99.5% ±0.2% | 0.502 |
| **XId Malignant melanomas** | 84.3%±3.1% | 91.6% ±1.7% | 0.032 | 92.2% ±1.3% | >0.9 | 95%±0.6% | 0.095 | 94.8% ±0.9% | >0.9 |

[1]The p-values were adjusted using the Holm-Bonferroni method, with comparison of survival rates of each study interval against its predecessor.

Abbreviations: AML, acute myeloid leukemias; GNB, ganglioneuroblastoma; ICCC, International Childhood Cancer Classification

(slope = 0.06, p <0.05) and soft-tissue malignancies (slope = 0.05, p <0.05). It should be noted that APC reflects the change over the study period and takes in consideration the rate of change (correlates with the slope) and the starting rates (denominator). The rates of hepatic tumors were very low in the first decade, and showed the highest APC, as mentioned above. The assessment pinpointed significant shifts for lymphomas and reticuloendothelial neoplasms, with critical junctures or joinpoints appearing in 2005 and 2014. The slope illustrated a decline of -0.06 from 1975–2005, an increase of 1.25 (p <0.05) from 2005–2014, and a subsequent decline of -0.92 from 2014–2019. An additional joinpoint was discerned for other

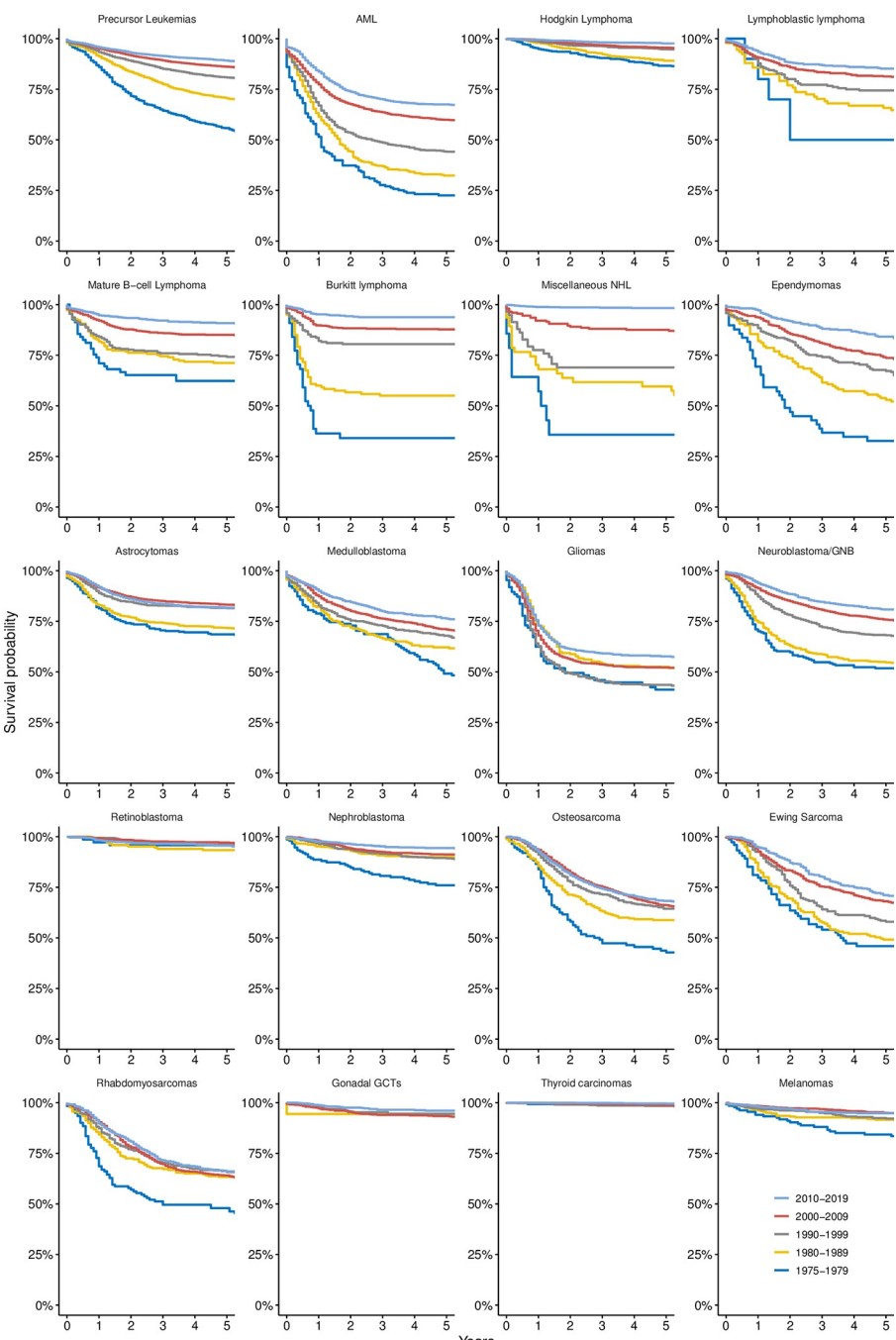

**Fig 5. Kaplan-Meier 5-year survival curves for the 20 most frequently observed diagnoses, stratified by the study interval.** Abbreviations: AML, acute myeloid leukemia; GCTs, germ cell tumors; GNB, ganglioneuroblastoma; NHL, non-Hodgkin lymphoma.

malignant epithelial neoplasms and malignant melanomas (Group XI) in 2005, as the slope shifted upwards from 0.14 (p <0.05) pre-2005 to 0.72* post-2005.

The 5-year relative survival analysis conducted through JoinPoint demonstrated gradual improvements over the years, with an AACS = 1.11 until 1988 and 0.42 post-1988 (Fig 9). Notably, survival for leukemias displayed significant improvement, reflected in an initial

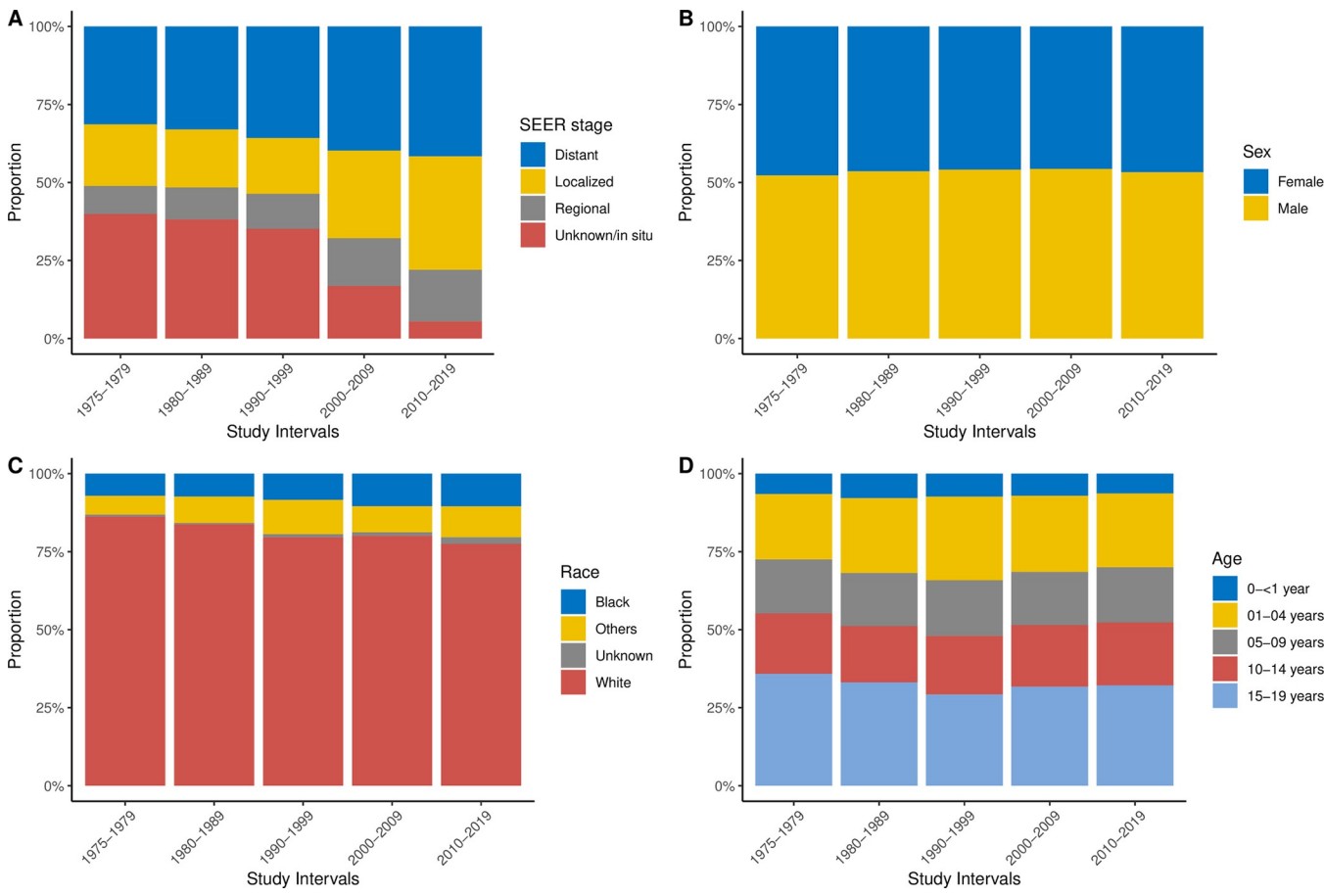

**Fig 6.** Proportions of (**A**) Seer stages, (**B**) sex, (**C**) race, and (**D**) age.

AACS of 5.15, followed by a lower yet steady rate of 0.90. Lymphomas exhibited an annual improvement of 0.58. In addition, survival of CNS malignancies and neuroblastoma also displayed enhancement at annual rates of 0.46 and 0.87, respectively. The survival for retinoblastoma remained relatively consistent, with a modest annual increase of 0.02, reflecting good overall survival. The survival of renal malignancies demonstrated a significant improvement at 3.19 until 1982 and a marginal increase of 0.09 afterwards. Hepatic malignancies showed an annual improvement of 1.2. Malignant bone tumors showed an increasing survival at 1.43 until 1992, followed by a decrease to 0.16.

The analysis of relative survival estimates by JoinPoint regression did not reveal any recent joinpoints. This observation underscores either the absence of substantial advancements in therapeutic strategies for these tumors in recent years or the maintenance of already high survival for many diseases.

## Discussion

Our study confirmed that cancer mortality has significantly decreased among pediatric patients. The annual age-standardized death rates dropped by almost half, from 4.9 to 2.3 per 100,000. However, these improvements in survival were not consistent across all diseases. Osteosarcoma and rhabdomyosarcoma did not consistently improve over the study period. Our Cox proportional hazards multivariable model revealed that the decade of diagnosis and

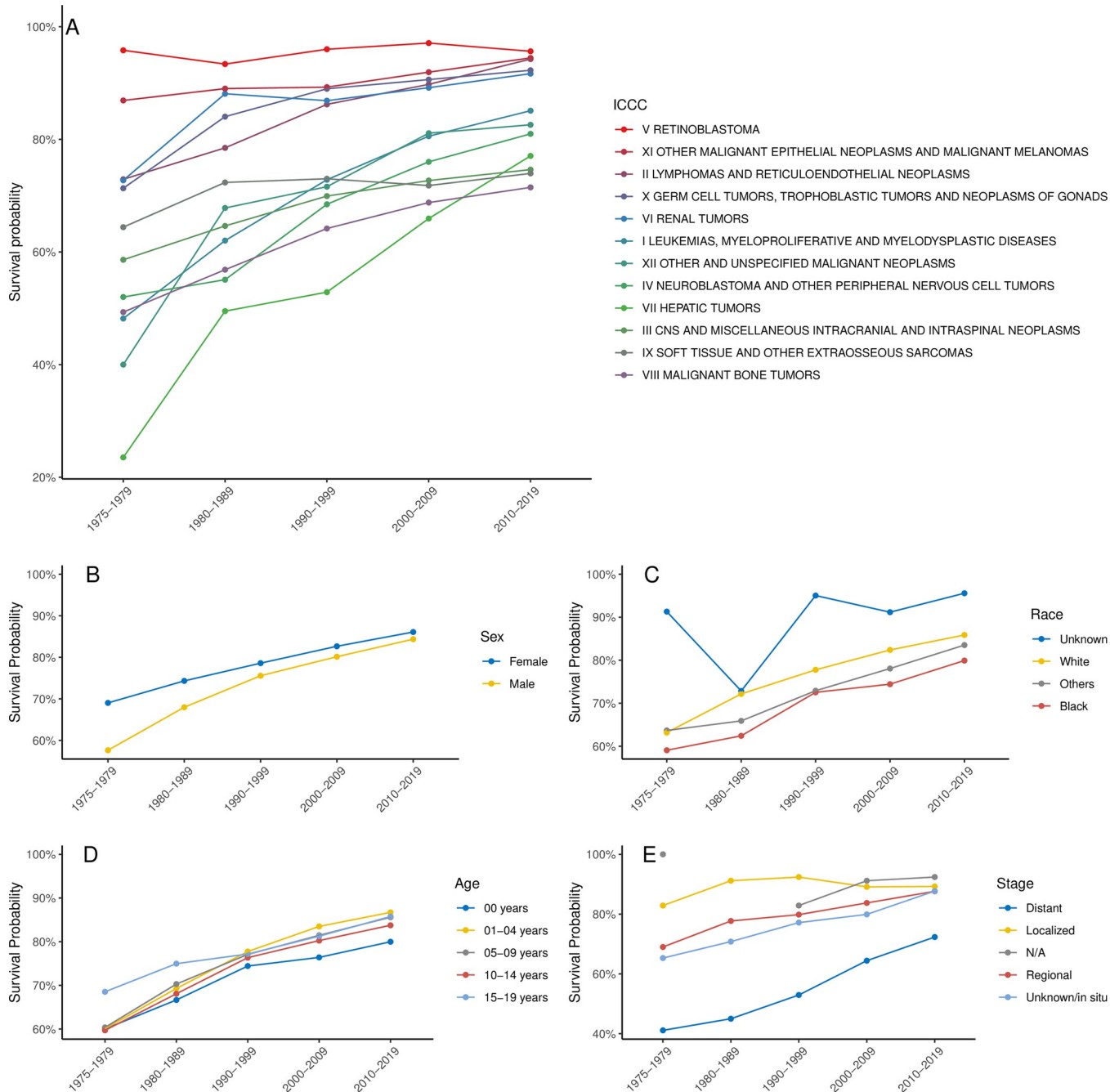

**Fig 7.** Survival trends stratified by (**A**) ICCC class, (**B**) sex, (**C**) race, (**D**) age groups, and (**E**) SEER stage.

the distant stage were the strongest predictors of outcome. Additionally, race and age were important predictors, with Black children experiencing worse survival, despite recent improvements and a narrowing of the gap.

The outcome of leukemias and lymphoma steadily improved. This is attributable to multiple changes in treatment and improvement of supportive care. Although the introduction of multiagent chemotherapy for acute lymphoblastic leukemia (ALL) and AML emerged in the 1950s and 1960s [14], this approach was further refined during the study period. Risk-based

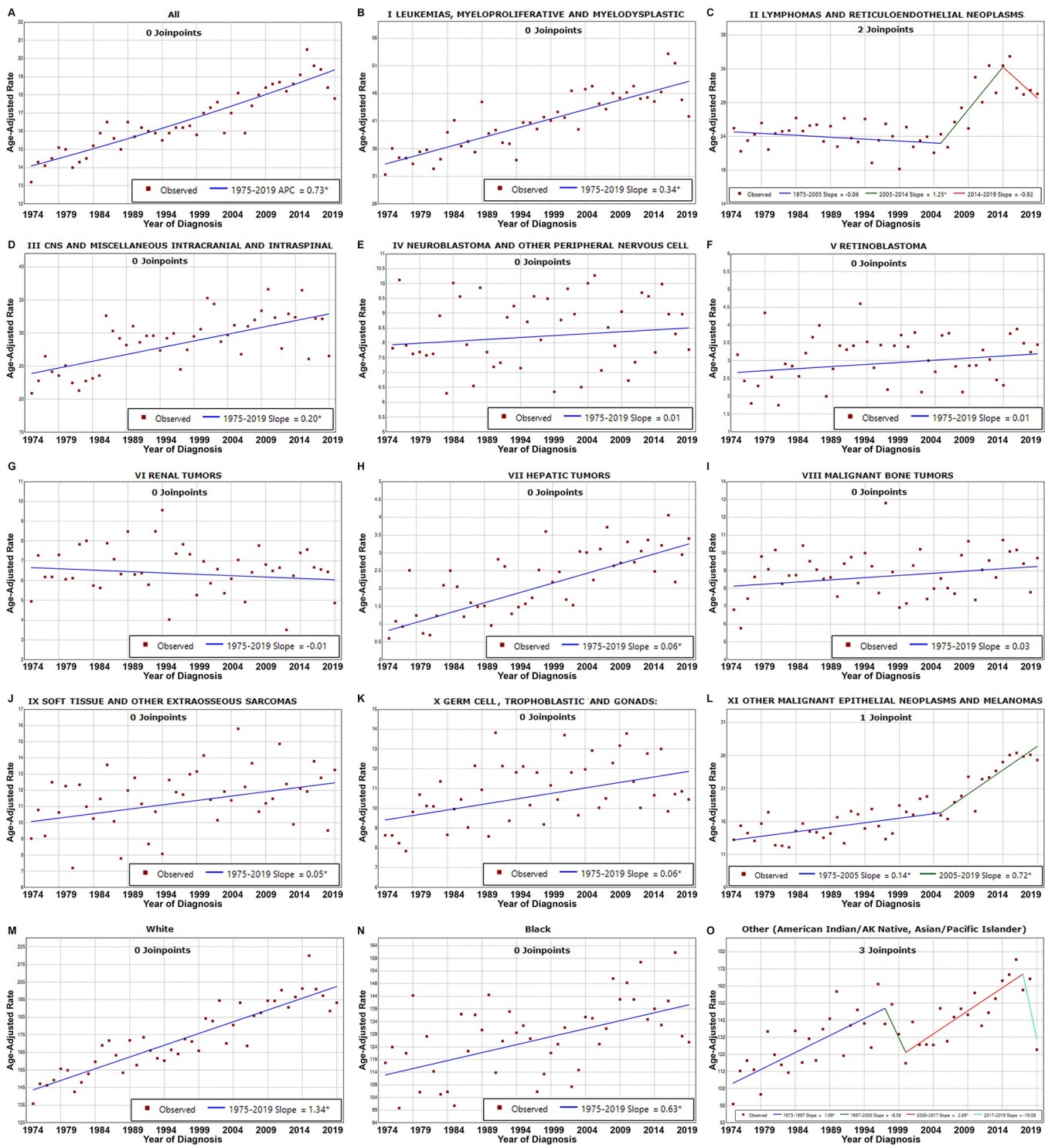

**Fig 8.** JoinPoint plots of the incidence of (**A**) all pediatric cancers, (**B-L**) different ICCC classes of pediatric cancer, and (**M-O**) all pediatric cancers by race, as reported by the SEER 8 data set (1975–2019).

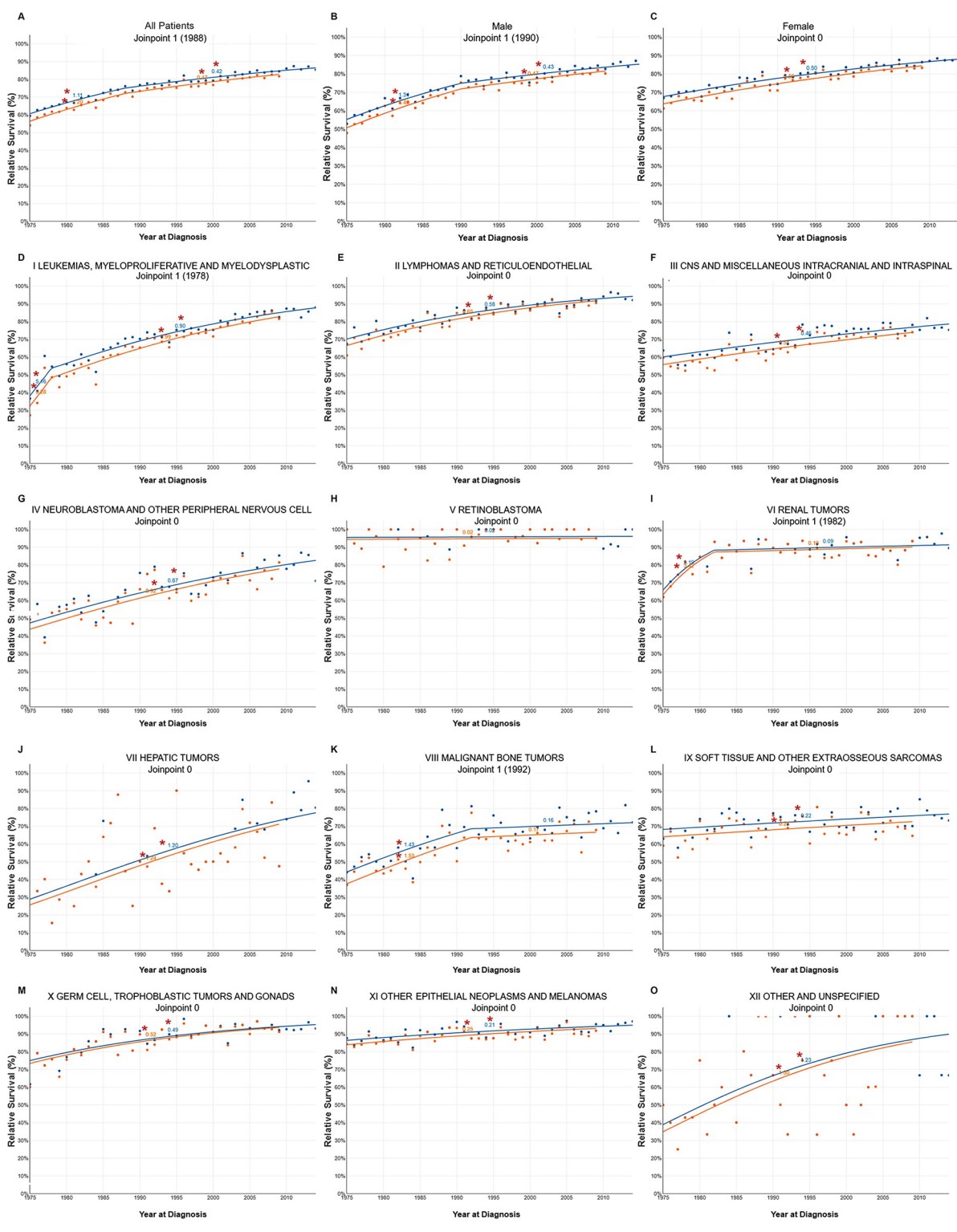

**Fig 9.** JoinPoint plots of the 5- and 10-year relative survival of (**A**) all children registered in SEER from 1975 to 2019, (**B**-**C**) grouped by sex or (**D**-**O**) ICCC class.

stratification was introduced in the 1980s [14], and minimal residual disease assessment was incorporated into treatment regimens the 1990s [15]. CNS prophylaxis was intensified using effective intrathecal chemotherapy, high-dose methotrexate was introduced [16] and tyrosine kinase inhibitors that improve the outcome of patients with Philadelphia chromosome–positive ALL were discovered in the 2000s [17]. Similarly, the outcome of AML improved after anthracycline-based induction therapy was introduced (1980s) [18], allogeneic hematopoietic stem cell transplantation was incorporated (1980s–1990s) [19], and the prognostic significance of certain genetic markers, such as *FLT3–ITD* and *NPM1*, was discovered (2000–2005) [20, 21].

The outcome of lymphomas improved during the study period, with the introduction of risk-stratified treatment for Hodgkin lymphoma [22, 23]; the introduction of the Lymphome Malin B protocol (late 1980s) improved the outcomes for pediatric patients with non-Hodgkin lymphoma, particularly those with Burkitt B-cell lymphoma or large-cell lymphomas [24] and the recent addition of rituximab to pediatric diffuse large B-cell lymphoma treatment regimens (2010s) [25].

The outcome of CNS tumors has dramatically improved due to risk-adapted therapy for medulloblastoma (1990s) [26]; the introduction of cisplatin, vincristine, and cyclophosphamide in the treatment of high-risk medulloblastoma [27]; and molecular subgrouping of medulloblastoma (2010s) [28]. Advances in surgical and radiotherapy techniques improved the outcome of children with ependymomas [29], and adding BRAF inhibitors (2013) improved the management of low-grade gliomas, though longer time is needed to appreciate the impact of this paradigm shift on outcome [30].

The National Wilms Tumor Study (NWTS) group led the early development of successful protocols for Wilms tumor, thereby dramatically improving patient outcomes. Ongoing international efforts in molecular stratification and fine-tuning of Wilms tumor therapy is further increasing survival [31].

The outcome of neuroblastoma, namely high-risk neuroblastoma, has dramatically improved after retinoids and high-dose chemotherapy with stem cell rescue were added to the treatment [32]. The recent incorporation of immunotherapy into high-risk neuroblastoma treatment regimens has also improved survival [32].

Rhabdomyosarcoma outlook improved with the introduction of effective chemotherapy and early locoregional control, but further improvements have proved to be very difficult, and survival of patients with intermediate- or high-risk disease remains suboptimal [33].

For osteosarcoma, introducing cisplatin-based chemotherapy in the 1980s greatly improved the outcome [34], but that of patients, particularly those with metastatic disease and poor response after neoadjuvant chemotherapy, remains suboptimal. For Ewing sarcoma, the combination of vincristine, doxorubicin, and cyclophosphamide, alternating with ifosfamide and etoposide with interval compression, has improved the outlook for patients with localized Ewing sarcoma [35].

From 1975 to 2019, the incidence of pediatric cancer has been on an upward trajectory, with a more pronounced increase in specific cancers, such as precursor cell leukemias (especially among white females) and AML. Other contributors to this trend include mature non-Burkitt B-cell lymphomas, CNS tumors (e.g., ependymomas and astrocytomas), hepatoblastoma, and malignant gonadal germinomas, the latter mainly increasing among the White population. Although the incidence of bone tumors has generally remained stable, osteosarcoma has shown an uptick among White individuals. Although advancements in diagnostic tests could partly explain these trends [36], the role of environmental factors cannot be dismissed and warrants in-depth epidemiologic scrutiny. For example, although increasing low-dose environmental radiation and increasing maternal age have been suggested as factors, those notions are difficult to prove and do not appear to explain the scale of rising incidence [37, 38]. A

previous report suggested there was an increase in childhood leukemia incidence between 1992 and 2004 but only a modest increase in overall childhood cancer incidence in the U.S. [39]. Similar trends have been observed in Australian studies, which also project a continuing rise in childhood cancer rates up to 2035 [40, 41]. The long-term data from the National Registry of Childhood Tumours further corroborate the rising incidence of childhood cancers in the U.K., emphasizing the necessity for ongoing research to understand the underlying factors [42]. Recent SEER data suggest a particular rise in early-onset cancers among adults younger than 50 years, especially among women (aged 30–39 years), while the incidence of cancer is declining in older individuals [43].

Our JoinPoint analysis revealed an interesting trend, with striking increase in incidence between 2004 and 2014, followed by a downward trend. This might reflect the interaction between a declining incidence for Hodgkin lymphomas and increasing incidence of non-Hodgkin (non-Burkitt) lymphomas. While this analysis has been a valuable tool in identifying trends in cancer incidence and survival, it does not fully account for external factors that may influence these trends. Public health interventions, advancements in medical technology, and shifts in healthcare policies can significantly affect the patterns observed. Future studies could benefit from integrating these external factors to provide a more comprehensive understanding of cancer trends and outcomes.

Lifestyle and environmental changes over recent decades, such as increased exposure to pollutants and carcinogens, may have contributed to the rising rates of pediatric cancers. For instance, certain pesticides and parental smoking have been linked to an increased risk of childhood leukemia [44–46]. Moreover, the prevalence of obesity and related metabolic changes might also play a role in this upward trend, with more evidence in young adults [47]. Genetic predispositions, coupled with these environmental factors, underline the complexity of cancer etiology, necessitating multifaceted approaches in both research and public health interventions to address this growing concern.

Racial disparities noted in our analysis is not a new finding. Even in standardized Children's Oncology Group trials where patients receive same regimens, black children had worse outcomes [48]. This is evident for different cancer types reported in previous analysis of SEER database, but not in patients treated at St. Jude Children's research hospital, where overlapping outcomes for all cancers were noted regardless of cancer type [49], suggesting that socioeconomic factors, rather than true biologic factors, are responsible for racial survival disparities.

There are several limitations to consider in our study. First, the SEER database from which our data were collected does not cover the entire United States. Additionally, older data were collected from the SEER 8 data set, which may not fully represent the current population and treatment practices. Second, we faced limitations in accessing crucial information about treatment modalities, such as specific chemotherapy regimens or surgical approaches due to the retrospective nature of the study. This limited our ability to thoroughly analyze the impact of different treatments on survival outcomes. Finally, although we observed changes in trends over time, we did not definitively identify the underlying causes of the fluctuations. Despite these limitations, our study provides valuable insights into pediatric cancer survival and highlights the need for further research to better understand the factors influencing these trends. Expanding this analysis to include other resources, e.g. Medicaid data, records of second malignancies, and deaths from other causes (e.g., cardiovascular) could provide a more comprehensive understanding of treatment-related toxicity. Integrating these additional data sources would offer a deeper insight into the long-term effects of cancer treatment in children, allowing for a broader analysis of how treatment impacts overall health and mortality.

## Supporting information

**S1 Table. Yearly trends in childhood cancer rates, stratified by race, age group, and sex, as derived from SEER\*stat trends analysis (Darker color indicates higher APC).**
(DOCX)

**S2 Table. Distribution and survival rates of childhood cancers by stage and decade according to the International Classification of Childhood Cancer (ICCC).**
(DOCX)

## Acknowledgments

The authors wish to extend their appreciation to Angela McArthur of St. Jude Children's Research Hospital for her invaluable assistance in editing the final manuscript.

## Author Contributions

**Conceptualization:** Iyad Sultan.

**Data curation:** Zeena Salman.

**Methodology:** Iyad Sultan, Yaseen Sultan.

**Project administration:** Ibrahim Qaddoumi.

**Resources:** Zeena Salman.

**Supervision:** Ibrahim Qaddoumi.

**Visualization:** Ahmad S. Alfaar, Yaseen Sultan.

**Writing – original draft:** Iyad Sultan.

**Writing – review & editing:** Ahmad S. Alfaar, Ibrahim Qaddoumi.

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
