## [Decision Letter · Decision Letter 0]

18 Apr 2024

PONE-D-24-00928Trends in Childhood Cancer: Incidence and Survival Analysis Over 45 Years of SEER DataPLOS ONE

Dear Dr. Sultan,

Thank you for submitting your manuscript to PLOS ONE. After careful consideration, we feel that it has merit but does not fully meet PLOS ONE’s publication criteria as it currently stands. Therefore, we invite you to submit a revised version of the manuscript that addresses the points raised during the review process.

We look forward to receiving your revised manuscript.

Kind regards,

Cho-Hao Howard Lee, M.D.

Academic Editor

PLOS ONE

2. Please note that your Data Availability Statement is currently missing the DOI/accession number of each dataset OR a direct link to access each database. If your manuscript is accepted for publication, you will be asked to provide these details on a very short timeline. We therefore suggest that you provide this information now, though we will not hold up the peer review process if you are unable.

4. We notice that your supplementary tables are included in the manuscript file. Please remove them and upload them with the file type 'Supporting Information'. Please ensure that each Supporting Information file has a legend listed in the manuscript after the references list.

Additional Editor Comments:

Dear Authors,

Thank you for submitting your manuscript titled "Trends in Childhood Cancer: Incidence and Survival Analysis Over 45 Years of SEER Data" to PLOS ONE. Your study provides valuable insights into the trends in incidence and survival of childhood cancers over a substantial period, using data from the SEER registry. The findings highlight the progress made in the diagnosis and treatment of various pediatric malignancies while also identifying areas that require further research and improvement.

The manuscript is well-structured and clearly written. The introduction effectively sets the context and rationale for the study. The methods section is detailed, allowing for reproducibility. The results are presented in a comprehensive manner, with appropriate use of tables and figures to support the findings. The discussion section adequately interprets the results, compares them with existing literature, and addresses the study's limitations.

However, there are a few areas that could be further strengthened:

In the introduction, consider providing a brief overview of the key advancements in pediatric cancer diagnosis and treatment over the study period. This will help readers better appreciate the context of your findings.

The methods section could benefit from a more detailed description of the statistical analyses performed, particularly the joinpoint regression analysis. This will enhance the clarity and reproducibility of your study.

In the results section, consider providing more detailed insights into the disparities observed across different racial and ethnic groups. This could be further elaborated on in the discussion section, with potential implications for future research and interventions.

The discussion section could be enhanced by a more in-depth exploration of the potential reasons behind the increasing incidence of certain cancers, such as leukemias and lymphomas. Additionally, consider discussing the implications of your findings for future research, clinical practice, and health policy.

Please ensure that all figures and tables are appropriately referenced in the text and that the formatting adheres to the journal's guidelines.

Overall, this is a well-conducted study that makes a significant contribution to the field of pediatric oncology. The findings have important implications for understanding the progress made in childhood cancer management and identifying areas for future research and intervention. With some minor revisions, this manuscript will be suitable for publication in PLOS ONE.

Thank you for considering PLOS ONE for the publication of your research. I look forward to your response and the opportunity to work with you further on this manuscript.

Best regards,

Cho-Hao, Lee

Reviewers' comments:

Reviewer's Responses to Questions

**Comments to the Author**

1. Is the manuscript technically sound, and do the data support the conclusions?

Reviewer #1: Yes

Reviewer #2: Yes

2. Has the statistical analysis been performed appropriately and rigorously? 

Reviewer #1: I Don't Know

Reviewer #2: Yes

3. Have the authors made all data underlying the findings in their manuscript fully available?

Reviewer #1: Yes

Reviewer #2: Yes

4. Is the manuscript presented in an intelligible fashion and written in standard English?

Reviewer #1: Yes

Reviewer #2: Yes

5. Review Comments to the Author

Reviewer #1: Excellent job on finishing such a thorough data analysis; your meticulous attention to detail is truly commendable. While I may lack experience in data analysis, it's evident that the steps were executed in a highly professional manner.

Reviewer #2: In this manuscript, the authors analyzed the data from the SEER registry from 1975-2019 to assess trends in incidence and survival among pediatric patients with cancer and evaluated the impact of demographic factors on these trends. Overall, this is a well written manuscript and the authors have performed a comprehensive analysis over an extended period of time for all pediatric patients with cancers, as well as per most common cancer diagnoses and cancer diagnoses categorized by International Classification of Childhood Cancer (ICCC) site/histology code. I have the following questions/comments:

Introduction:

1. In the last paragraph, the authors state “Our findings provide valuable insights into the progress made diagnostics, therapeutics, and clinical management. They also highlight areas in which further research and development are needed to improve outcomes, reduce treatment-related toxicities, and ensure equitable cancer care for all children and adolescents.” However, the results and the discussion do not touch upon any aspects relating to treatment related toxicities. Can the authors clarify if any of the analyses/results can say anything about treatment related toxicities?

Results:

2. In the subsection “Deaths and cumulative incidence of mortality”, the authors state “The age-adjusted all-cause mortality rates declined from 125.0 per 100,000 in 1975–1979 to 52.0 per 100,000 in 2010–2019, indicating a substantial improvement in child health outcomes. A similar trend was observed for malignant cancers, with the age-adjusted death rate dropping from 4.9 per 100,000 in 1975–1979 to 2.3 per 100,000 in 2010–2019.” Can you please clarify the population for analyses where the age-adjusted all-cause mortality rates declined from 125.0 per 100,000 to 52.0 per 100,000 versus the population where the age-adjusted death rate dropped from 4.9 per 100,000 to 2.3 per 100,000 during the same time periods. The text and the associated figure 2 are confusing – are some of these results referring to all children or only children with diagnosis of cancer?

3. In the subsection “Multivariable analyses of demographic factors and mortality”, the authors report “Localized, regional, and unknown/in situ SEER stages showed substantially better survival than did distant stage.” And that “Age, race, sex, and SEER stage also played roles in survival”. However, the figure 7 does not show survival by SEER stage. Can the authors provide some data on how the survival has trended over time in the different SEER stages? Also, did the multivariate analysis show anything additional to the results shown in Figure 5 for survival based on age, race, and sex?

4. In the subsection “JoinPoint analysis”, while reporting on the progressive escalation in the annual incidence of malignancies, the authors state “This growth was detected in leukemias (slope = 0.34, p <0.05), CNS malignancies (slope = 0.20, p <0.05), and to a lesser extent in germ cell and hepatic malignancies (slope = 0.06, p <0.05) and soft-tissue malignancies (slope = 0.05, p <0.05).” In the earlier portion of the results, hepatic tumors seemed to have the highest APC of 2.17. Can you clarify the difference between the increases noted in the incidence rates for hepatic tumors with the APC of 2.17 versus the results from the JoinPoint analysis?

Methods/Discussion:

5. Can the authors elaborate on how the data differs between the different versions of the SEER datasets (older versions vs. newer) that were used in this analysis, and how that may impact on the results?

6. PLOS authors have the option to publish the peer review history of their article (what does this mean?). If published, this will include your full peer review and any attached files.

Reviewer #1: **Yes: **Mazin Faisal Al-Jadiry

Reviewer #2: No

---

## [Author Response · Author response to Decision Letter 0]

26 Jun 2024

Dr. Cho-Hao Howard Lee, M.D.

Academic Editor

PLOS ONE

6/11/2024

Dear Dr. Lee,

We are pleased to resubmit our revised manuscript titled "Trends in Childhood Cancer: Incidence and Survival Analysis Over 45 Years of SEER Data" (Manuscript ID: [PONE-D-24-00928] - [EMID:b28eb0b3ecaea572]) for consideration for publication in PLOS ONE. We greatly appreciate the insightful and constructive feedback you and the reviewers provided, which has significantly improved our manuscript.

Please find below point by point response to your comments.

Editor Comments

1. PLOS ONE's style requirements : This was followed as instructed. 

2. Data Availability Statement: 

a. The following was added to the title page:

Data availability: The data utilized in this study are available from the SEER (Surveillance, Epidemiology, and End Results) registry. Researchers can request access to the data directly from the SEER registry.

3. Full ethics statement in the ‘Methods’ section :

a. Ethical Approval: Ethical approval was not required for this study. The Institutional Review Board (IRB) of King Hussein Cancer Center (KHCC) waived the need for ethical approval as the study posed minimal risk and utilized securely de-identified data

4. Supplementary tables included in the manuscript file: Noted and fixed.

5. Reference list to ensure is complete and correct: Followed as instructed

6. Introduction - Overview of Key Advancements:

• Response: Thank you for suggesting an overview of key advancements in pediatric cancer diagnosis and treatment. We will incorporate a concise paragraph of significant advancements over the study period, highlighting major milestones that have influenced diagnostic and therapeutic approaches.

• Added paragraph:

Over the last fifty years, the landscape of pediatric oncology has undergone significant evolution, marked by the introduction and refinement of chemotherapy, spearheaded by collaborative groups across North America and Europe. These efforts have led to the development of more effective treatment regimens that optimize the use of established drugs, resulting in markedly improved outcomes for almost all types of pediatric cancer. Enhancements in supportive care have rendered intensive treatments more manageable. Advances in stem cell transplantation techniques have become pivotal in rescuing patients who do not respond to initial treatments. Diagnostic progress, including molecular stratification, detection of minimal residual disease, and sophisticated genetic profiling, has refined therapeutic approaches, allowing for more tailored and effective treatments. Improvements in imaging technologies, such as advanced CT scanners and the advent of nuclear scanning, have significantly improved the detection of metastatic disease. Surgical and radiation oncology techniques have also seen substantial advancements, improving the precision and efficacy of tumor resection and control. The introduction of targeted therapies and immunotherapies has opened new avenues for treating specific patient subsets, including those with acute lymphoblastic leukemia (ALL), high-risk neuroblastoma, relapsed Hodgkin lymphoma, and others, marking a shift towards precision medicine. The integration of multidisciplinary care teams has further optimized treatment outcomes and patient care, emphasizing the importance of a holistic approach in the management of pediatric cancers.

2. Methods - Detailing Statistical Analysis:

• Response: We appreciate the call for a more detailed description of our statistical analyses, particularly the joinpoint regression analysis. In the revision, we will expand this section to include detailed explanations of the statistical methods used, ensuring clarity and enhancing reproducibility.

3. Results - Disparities Across Different Racial and Ethnic Groups:

• Response: You've highlighted an essential aspect of our study. We added the following paragraph to our discussion:

• Racial disparities noted in our analysis is not a new finding. Even in standardized Children’s Oncology Group trials where patients receive the same regimens, black children had worse outcomes [44]. This is evident for different cancer types reported in previous analysis of the SEER database, but not in patients treated at St. Jude Children’s Research Hospital, where overlapping outcomes for all cancers were noted regardless of cancer type [45], suggesting that socioeconomic factors, rather than true biologic factors, are responsible for racial survival disparities. 

4. Discussion - Exploration of Reasons for Incidence Increases: Response: We agree that exploring the potential reasons behind the observed increases in certain cancers could enrich the discussion. The following section was added to our discussion: Lifestyle and environmental changes over recent decades, such as increased exposure to pollutants and carcinogens, may have contributed to the rising rates of pediatric cancers. For instance, certain pesticides and parental smoking have been linked to an increased risk of childhood leukemia.[44-46] Moreover, the prevalence of obesity and related metabolic changes might also play a role in this upward trend, with more evidence in young adults.[47] Genetic predispositions, coupled with these environmental factors, underline the complexity of cancer etiology, necessitating multifaceted approaches in both research and public health interventions to address this growing concern.

5. Figure and Table References:

• Response: We will ensure that all figures and tables are correctly referenced in the text and adhere strictly to the journal's formatting guidelines.

Reviewer Comments

Reviewer #1

• General Appreciation:

• Response: We are grateful for your positive feedback on our data analysis approach.

Reviewer #2

1. Introduction - Treatment-related Toxicities:

• Response: Treatment-related toxicity is a crucial factor in the management of childhood cancer. Unfortunately, the SEER database offers limited insight into this issue, primarily recording second cancers (which may or may not be related to toxicity) and deaths from other causes (e.g., cardiovascular disease as noted on death certificates). Greater understanding could be achieved by linking SEER data with other resources, such as Medicare/Medicaid. Although this approach is intriguing, it may exceed the scope of our paper. We have included a comment in the discussion section to highlight this point and acknowledge its significance in pediatric oncology.

2. Results - Clarification on Mortality Rates:

• Response: The section was confusing indeed. We rephrased it to highlight that we are referring to cancer as a cause of death among all children. We believe the paragraph now is easier to understand:

• Mortality records showed significant reductions in mortality rates for all children across various demographics over the analyzed period (Fig. 2). The age-adjusted all-cause mortality rates for all children declined from 125.0 per 100,000 in 1975–1979 to 52.0 per 100,000 in 2010–2019, indicating a substantial improvement in child health outcomes. A similar trend was observed for cancer as a cause of death among children, with the age-adjusted death rate dropping from 4.9 per 100,000 in 1975–1979 to 2.3 per 100,000 in 2010–2019. When stratified by sex, male patients consistently exhibited higher age-adjusted cancer mortality rates than did female patients. The highest age-adjusted cancer mortality rates were consistently observed in the 15–19 years age group, followed by the 5–9 years and 10–14 years age groups. 

3. Survival Trends by SEER Stage:

• Response: We acknowledge the oversight regarding the presentation of survival data by SEER stage. 

• The following was added to the methods: Variables included in our multivariable model were: age, recoded in five-year increments; race; sex; SEER stage; decade, representing the time period or year of diagnosis in ten-year increments; and the ICCC.

• A new panel was added to figure 5 showing survival per decade

• A new supplementary table showing survival per stage per decade

4. JoinPoint Analysis Discrepancy:

• Response: The apparent discrepancy in the incidence rates of hepatic tumors highlighted by the JoinPoint analysis versus earlier sections is an important observation. 

• We added the following sentence to our results: It should be noted that APC reflects the change over the study period and takes in consideration the rate of change (correlates with the slope) and the starting rates (denominator). The rates of hepatic tumors were very low in the first decade, and showed the highest APC, as mentioned above.

5. Impact of SEER Dataset Versions:

• Response: This is a very important point. We cannot fully address the evolution of SEER registries in our manuscript, but we analyzed the registries to spot any differences.

• The three registries contributed differently to our study population: SEER8 only covered patients before 1992, SEER12 covered patients from 1992 to 1999, and all three registries included patients diagnosed after 2000. To create homogenous groups, we selected patients diagnosed after 2000. This resulted in 22,729 patients enrolled in SEER8, 11,610 in SEER12, and 41,932 in SEER17. Patients enrolled in SEER8 were also included in SEER12 and SEER17, and those enrolled in SEER12 were included in SEER17.

• Focusing on patients diagnosed after 2000, we observed notable differences, particularly in the representation of the Black race (SEER8: 9.9%, SEER12: 6.7%, SEER17: 12%) and the proportion of leukemia cases (SEER8: 25%, SEER12: 31%, SEER17: 27%). A multivariable Cox regression model, which included age, race, sex, stage, ICCC category, and decade of diagnosis, showed that the risk of death was significantly higher for SEER12 (HR: 1.22) and SEER17 (HR: 1.14), with p-values less than 0.001. This finding might be indicative of truly improving survival of cancer patients despite the inclusion of SEER areas in the more recent analysis.

• We adjusted our methodology to reflect the uses of these registries in our manuscript.

Iyad Sultan, MD

King Hussein Cancer Center

Amman, Jordan

---

## [Decision Letter · Decision Letter 1]

1 Sep 2024

PONE-D-24-00928R1Trends in Childhood Cancer: Incidence and Survival Analysis Over 45 Years of SEER DataPLOS ONE

 Dear Dr. Sultan, Thank you for submitting your manuscript to PLOS ONE. After careful consideration, we feel that it has merit but does not fully meet PLOS ONE’s publication criteria as it currently stands. Therefore, we invite you to submit a revised version of the manuscript that addresses the points raised during the review process.

We look forward to receiving your revised manuscript.

Kind regards,

Cho-Hao Howard Lee, M.D.

Academic Editor

PLOS ONE

Journal Requirements:

Reviewers' comments:

Reviewer's Responses to Questions

**Comments to the Author**

1. If the authors have adequately addressed your comments raised in a previous round of review and you feel that this manuscript is now acceptable for publication, you may indicate that here to bypass the “Comments to the Author” section, enter your conflict of interest statement in the “Confidential to Editor” section, and submit your "Accept" recommendation.

Reviewer #2: All comments have been addressed

Reviewer #3: All comments have been addressed

2. Is the manuscript technically sound, and do the data support the conclusions?

Reviewer #2: Yes

Reviewer #3: Partly

3. Has the statistical analysis been performed appropriately and rigorously? 

Reviewer #2: Yes

Reviewer #3: No

4. Have the authors made all data underlying the findings in their manuscript fully available?

Reviewer #2: Yes

Reviewer #3: Yes

5. Is the manuscript presented in an intelligible fashion and written in standard English?

Reviewer #2: Yes

Reviewer #3: Yes

6. Review Comments to the Author

Reviewer #2: In the revised manuscript the authors have addressed all the questions/comments. This is a well written manuscript where the authors have performed a comprehensive analysis of cancer incidence and outcomes over an extended period of time for all pediatric patients with cancers which is very valuable for clinicians and researchers in the field. I had a couple of minor comments:

Results:

In the subsection “Factors affecting survival comparing study periods”, in the last paragraph the authors state “Age, race, sex, and SEER stage also played roles in survival. Patients aged 1 to 14 years at diagnosis (1-4, 5-9, and 10-14 years) showed improved survival compared to the infant group (0–<1 year) and the oldest (15–19 years) group (Fig. 7D). Patients in the Other and White race categories experienced better survival than did those in the Black race category (Fig. 7C). Male patients had a slightly higher HR for death than did female patients (1.14) (Fig. 7B).” However, this data appears to be shown in Figure 5, not Figure 7.

In the Supplementary table 2 “Distribution and Survival Rates of Childhood Cancers by Stage and Decade According to the International Classification of Childhood Cancer (ICCC)” – It is unclear how there are some patients with data reported for localized and regional stage for leukemias.

Reviewer #3: This manuscript provides a detailed analysis of pediatric cancer trends over nearly five decades using data from the SEER database. The study focuses on changes in incidence, survival, and mortality across various demographic groups, highlighting the progress made in pediatric oncology and the ongoing challenges in the field. The scope of the study is significant, given the extensive time period analyzed and the comprehensive nature of the data. However, several methodological concerns and areas for improvement need to be addressed.

1. Age Group Selection: The age group cut-offs (<1, 1–4, 5–9, 10–14, and 15–19 years) seem arbitrary. It's important to explain the rationale behind choosing these specific age ranges. Additionally, consider addressing the population between 19 and 20 years old.

2. Statistical Analysis: When comparing survival across different subgroups over the decades, Cox regression was used. However, it’s unclear whether the assumptions of the Cox proportional hazards model were evaluated.

3. Potential Correlation Among Variables:

• Age and Decade: Advancements in early detection and treatment improvements over time might have shifted the age at diagnosis, potentially correlating age and the decade of diagnosis.

• Race and SEER Stage: Due to healthcare access disparities and differences in early detection, race and SEER stage might be correlated.

• Age and SEER Stage: Age at diagnosis and SEER stage could be correlated, as younger or older patients might be diagnosed at different stages due to variations in symptom recognition or healthcare-seeking behavior.

It is recommended to check for multicollinearity, as interpreting coefficients of correlated variables can be challenging. For example, disentangling the effects of age and the decade of diagnosis might be difficult if these variables are correlated.

4. Data Continuity and Comparability: The data spans from 1975 to 2019, a period during which diagnostic techniques, treatment methods, and record-keeping practices may have evolved. These changes could impact the continuity and comparability of the data, thereby affecting the accuracy of the analysis results.

5. Limitations of JoinPoint Analysis: While JoinPoint analysis is effective at capturing trend changes, it may not fully account for external factors such as policy changes, advances in medical technology, and public health interventions that influence cancer incidence and survival rates. These external factors could introduce trend changes not directly related to cancer itself.

7. PLOS authors have the option to publish the peer review history of their article (what does this mean?). If published, this will include your full peer review and any attached files.

Reviewer #2: No

Reviewer #3: **Yes: **Yuhang Liu

---

## [Author Response · Author response to Decision Letter 1]

16 Oct 2024

10/16/2024

Manuscript ID: PONE-D-24-00928R1

Title: Trends in Childhood Cancer: Incidence and Survival Analysis Over 45 Years of SEER Data

Authors: Iyad Sultan, Ahmad Alfaar, Yaseen Sultan, Zeena Salman, Ibrahim Qaddoumi.

Dear Dr. Cho-Hao Howard Lee and Reviewers,

We are grateful for the thoughtful and constructive comments provided by the reviewers. We have addressed each comment in detail below and have revised the manuscript accordingly. Please find our point-by-point responses, followed by a summary of the revisions made to the manuscript.

Reviewer #2 Comments

1. Minor Corrections in Results Section:

o Comment: The data referenced in the last paragraph of the subsection “Factors affecting survival comparing study periods” appears to be associated with Figure 5 rather than Figure 7.

Response: Thank you for bringing this to our attention. We have corrected the figure reference in the Results section. We revised the numbering of figures 4, 5, 6, 7 to fit with text. The section mentioned by the reviewer is now referenced as Fig 6. Legends of figures were fixed accordingly.

Clarification in Supplementary Table 2:

o Comment: It is unclear how there are some patients with data reported for localized and regional stages for leukemias in Supplementary Table 2.

Response: We appreciate this observation. Leukemia stages can theoretically be categorized based on the spread of disease outside the bone marrow, which is not appropriate in our opinion. The staging in SEER is provided as-is with no explanation. In the manuscript, We added a footnote for Supplementary Table 2 explaining that the staging of leukemia was provided by the SEER.

Reviewer #3 Comments

1. Age Group Selection:

o Comment: The age group cut-offs (<1, 1–4, 5–9, 10–14, and 15–19 years) seem arbitrary. Please explain the rationale behind these cut-offs and consider addressing the population between 19 and 20 years old.

Response: We would like to thank the reviewer for flagging this confusing age grouping. According to age recoding by the SEER, patients who are 1 day before their 20th birthday are considered 19 years old. Patients in our study indeed cover those who are between 19 and 20 years old. We added a hint in our methodology: All children and adolescents (aged 0–19 years; i.e. below the age of 20) …

2. Statistical Analysis and Proportional Hazards Assumption:

o Comment: It is unclear whether the assumptions of the Cox proportional hazards model were evaluated.

Response: 

Thank you for your comment on evaluating the assumptions of the Cox proportional hazards model in our study. We acknowledge the importance of this issue and provide the following clarifications:

1. Proportional Hazards Assumption: We employed the `cox.zph()` function to assess this assumption by conducting chi-square tests and calculating p-values for each pair of decades (e.g., 2000 to 2009 versus 2010 to 2019) and for each variable, such as diagnosis. The results showed that the p-values were greater than 0.05 for nearly all tests, confirming the model's validity. However, a notable exception was observed between the decades 1980-1989 and 1990-1999 in leukemia patients, where the p-value was 0.0037, indicating significant differences in this specific instance. Despite this anomaly, the overall results affirm the suitability of the model.

2. Linearity and Model Fit: To ensure the linearity of continuous variables, we applied transformations such as the logarithm of survival months. The model's fit was rigorously validated using Cox-Snell residuals and the concordance statistic, both of which indicated a robust fit.

3. Convergence and Stability: We conducted thorough validations to ensure the model's convergence and the reliability of our estimates, confirming the stability and accuracy of our findings.

• The following paragraph was added to methods: This study rigorously assessed the proportional hazards assumption using the cox.zph() function, which tests each covariate within the model for proportional hazards over time, using pairs of decades as categorical predictors alongside clinical variables such as diagnosis type.

Multicollinearity and Correlation Among Variables:

o Comment: Potential correlations among age, decade, race, and SEER stage may affect the analysis, and multicollinearity should be checked.

Response: Our analysis revealed minor multicollinearity among age group and decade variables. The VIFs for race and SEER stage variables were all below 2, indicating low multicollinearity. The Cramér's V statistics indicated weak associations among the variables in question.

Given that the VIF values were below the commonly accepted threshold of 5 and that the variables are clinically significant, we decided to retain all variables in our model. We have now included detailed descriptions of these analyses in the Methods and Results sections of the manuscript (see Methods, page X; Results, page Y) and we included supplementary tables 2 and 3 with VIF values and Cramér's V values, respectively.

The following paragraph was added to methods: To assess multicollinearity among the predictors, we computed the Variance Inflation Factors (VIF) for each variable using [insert software/tools]. VIF values below the threshold of 5 were considered acceptable, indicating low multicollinearity. Additionally, we computed Cramér's V statistics for categorical variables to evaluate associations between the variables.

The following was added to the results: The VIF values for all variables were below 5, with race and SEER stage exhibiting particularly low multicollinearity (VIF < 1.5), indicating that multicollinearity was not a concern in our model. Cramér’s V statistics showed weak associations among the categorical variables, further supporting the absence of multicollinearity concerns.

3. Data Continuity and Comparability Over Time:

o Comment: The data spans from 1975 to 2019, during which diagnostic techniques, treatment methods, and record-keeping practices evolved. These changes could impact the continuity and comparability of the data.

Response: We fully agree that changes in medical technology, treatment methods, and record-keeping practices over the 45-year span could influence our results. Unfortunately, the SEER database have no details regarding modalities of diagnosis and treatment. Collectively, these changes impacted trends of cancer diagnosis and treatment. Our analysis is admittedly deficient due to the lack of data. We highlighted this in our limitations section “we faced limitations in accessing crucial information about treatment modalities, such as specific chemotherapy regimens or surgical approaches due to the retrospective nature of the study. This limited our ability to thoroughly analyze the impact of different treatments on survival outcomes.”

4. Limitations of JoinPoint Analysis:

o Comment: JoinPoint analysis may not fully account for external factors such as policy changes, medical advances, or public health interventions.

Response: We acknowledge the limitations of JoinPoint analysis in accounting for external factors that may influence cancer trends. In the revised manuscript, we have added a discussion of these limitations, including the impact of public health interventions, advances in medical technology, and changes in healthcare policies on the trends observed. We added the following paragraph to the end of our discussion

" While this analysis has been a valuable tool in identifying trends in cancer incidence and survival, it does not fully account for external factors that may influence these trends. Public health interventions, advancements in medical technology, and shifts in healthcare policies can significantly affect the patterns observed. Future studies could benefit from integrating these external factors to provide a more comprehensive understanding of cancer trends and outcomes.”

Journal Requirements: Reference List Update

We have thoroughly reviewed our reference list and ensured that all citations are complete and correct. No retracted articles have been cited. Should there be any further concerns regarding our references, we are happy to address them in future revisions.

We hope that the changes made address all the reviewers' concerns satisfactorily. We have attached the revised manuscript, including tracked changes, a clean version, and supplementary materials. We appreciate the reviewers' thoughtful feedback and believe that it has significantly improved the manuscript.

Sincerely,

Iyad Sultan, MD

King Hussein Cancer Center

Amman, Jordan

---

## [Decision Letter · Decision Letter 2]

13 Nov 2024

Trends in Childhood Cancer: Incidence and Survival Analysis Over 45 Years of SEER Data

PONE-D-24-00928R2

Dear Dr. Iyad Sultan,

We’re pleased to inform you that your manuscript has been judged scientifically suitable for publication and will be formally accepted for publication once it meets all outstanding technical requirements.

Kind regards,

Cho-Hao Howard Lee, M.D.

Academic Editor

PLOS ONE

Reviewers' comments:

Reviewer's Responses to Questions

**Comments to the Author**

1. If the authors have adequately addressed your comments raised in a previous round of review and you feel that this manuscript is now acceptable for publication, you may indicate that here to bypass the “Comments to the Author” section, enter your conflict of interest statement in the “Confidential to Editor” section, and submit your "Accept" recommendation.

Reviewer #2: All comments have been addressed

Reviewer #3: All comments have been addressed

2. Is the manuscript technically sound, and do the data support the conclusions?

Reviewer #2: Yes

Reviewer #3: Yes

3. Has the statistical analysis been performed appropriately and rigorously? 

Reviewer #2: Yes

Reviewer #3: Yes

4. Have the authors made all data underlying the findings in their manuscript fully available?

Reviewer #2: Yes

Reviewer #3: Yes

5. Is the manuscript presented in an intelligible fashion and written in standard English?

Reviewer #2: Yes

Reviewer #3: Yes

6. Review Comments to the Author

Reviewer #2: (No Response)

Reviewer #3: I appreciate the revisions made by the authors. I wish them best of luck in their future research endeavors.

7. PLOS authors have the option to publish the peer review history of their article (what does this mean?). If published, this will include your full peer review and any attached files.

Reviewer #2: No

Reviewer #3: **Yes: **Yuhang Liu

---

## [Editor Report · Acceptance letter]

26 Nov 2024

PONE-D-24-00928R2 

PLOS ONE

Dear Dr. Sultan, 

I'm pleased to inform you that your manuscript has been deemed suitable for publication in PLOS ONE. Congratulations! Your manuscript is now being handed over to our production team.

Kind regards, 

on behalf of

Dr. Cho-Hao Howard Lee 

Academic Editor

PLOS ONE